# Global meta-analysis shows pervasive phosphorus limitation of aboveground plant production in natural terrestrial ecosystems

Enqing Hou [1,2,3]*, Yiqi Luo[3], Yuanwen Kuang[1,2], Chengrong Chen [4], Xiankai Lu[1,2], Lifen Jiang[3], Xianzhen Luo[1,2] & Dazhi Wen[1,2]*

Phosphorus (P) limitation of aboveground plant production is usually assumed to occur in tropical regions but rarely elsewhere. Here we report that such P limitation is more widespread and much stronger than previously estimated. In our global meta-analysis, almost half (46.2%) of 652 P-addition field experiments reveal a significant P limitation on aboveground plant production. Globally, P additions increase aboveground plant production by 34.9% in natural terrestrial ecosystems, which is 7.0–15.9% higher than previously suggested. In croplands, by contrast, P additions increase aboveground plant production by only 13.9%, probably because of historical fertilizations. The magnitude of P limitation also differs among climate zones and regions, and is driven by climate, ecosystem properties, and fertilization regimes. In addition to confirming that P limitation is widespread in tropical regions, our study demonstrates that P limitation often occurs in other regions. This suggests that previous studies have underestimated the importance of altered P supply on aboveground plant production in natural terrestrial ecosystems.

[1] Key Laboratory of Vegetation Restoration and Management of Degraded Ecosystems, South China Botanical Garden, Chinese Academy of Sciences, Guangzhou 510650, China. [2] Center of Plant Ecology, Core Botanical Gardens, Chinese Academy of Sciences, Guangzhou 510650, China. [3] Center for Ecosystem Science and Society, Northern Arizona University, Flagstaff, AZ 86011, USA. [4] Australian Rivers Institute, School of Environment and Science, Griffith University, Nathan, QLD 4111, Australia. *email: houeq@scbg.ac.cn; dzwen@scbg.ac.cn

Nutrient limitation of aboveground plant production has been widely acknowledged[1–5]. In terrestrial ecosystems, nitrogen (N) has been considered as the most important limiting nutrient of aboveground plant production[3,6]; phosphorus (P) has also been viewed as important, but mainly in lowland tropical regions where soils are generally strongly weathered[4,7]. This prevalent view, however, has been challenged by an increasing number of significant P limitation cases in areas other than the lowland tropical regions (e.g., tundra regions)[1,8,9]. For example, significant P limitation on aboveground plant production has also been found in some temperate areas with strongly weathered soils[10–12]. Despite the important role of P in aboveground plant production, we still lack a clear understanding of where, to what degree, and under what conditions P limits aboveground plant production over the global land surface[5,8,13]. As a consequence, none of the tens of models in the fifth phase of the Coupled Model Intercomparison Project (CMIP5) archive represents terrestrial P biogeochemistry, which causes substantial uncertainty in estimates of strength of the terrestrial carbon (C) sink through the 21st century[13,14].

Here we report the distribution, magnitude, and drivers of P limitation of aboveground plant production in terrestrial ecosystems worldwide. To accomplish this, we use a global database of 652 P-addition field experiments compiled from 285 papers published between 1955 and 2017 (Supplementary Figs. 1a and 2). The database includes P-addition experiments in all major types of terrestrial ecosystems, including both natural terrestrial ecosystems (436 experiments in forests, grasslands, tundras, or wetlands) and croplands (216 experiments) (Supplementary Tables 1 and 2). The number of P-addition experiments in natural terrestrial ecosystems in this study is 3.8–8.8 times greater than the number in the previous meta-analyses ($N = 50–117$)[1,8,9,15]. In addition, 41.7% of the experiments in our database were published after 2007 and few of these were included in previous syntheses dedicated to N–P interactions (Supplementary Fig. 2). The collected experiments are located on all continents except Antarctica (Supplementary Fig. 1a) and have wide ranges of mean annual precipitation (MAP, 80–5302 mm $yr^{-1}$) and mean annual temperature (MAT, −12.1 to 27.5 °C) (Supplementary Table 1). Compared to previous datasets, this up-to-date dataset better captures Earth's diverse terrestrial habitats and thereby provides a much clearer understanding of the role of P supply on aboveground plant production.

To explore the global distribution of P limitation, we first estimate a threshold value of P limitation, i.e., a critical P effect size that best corresponds to a critical z-score at $P = 0.05$, based on the statistical results provided in the 285 papers that comprised our database (see "Methods" section; Supplementary Fig. 3). We then map the global distribution of significant and non-significant P limitation cases. We quantify the magnitude of P limitation at the global scale as well as in various groups of ecosystems using a meta-analysis approach typically used in ecological studies, i.e., the natural logarithm transformed response ratio (Ln($RR$)) of aboveground plant production to P additions weighted by the inverse variance (details in "Methods" section)[16–19]. Finally, we explore the effects of climate, ecosystem properties, and fertilization regimes and their relative importance in predicting the P effect size using a boosted regression tree method[20]. In general, we show a more widespread and much stronger P limitation of aboveground plant production in natural terrestrial ecosystems than previously suggested[1,8,9,15].

## Results and discussion

**Globally distributed P limitation**. Our synthesis revealed that P limitation of aboveground plant production is globally distributed,

from tropical to arctic regions, spanning over 131° in latitude (54.8°S–76.5°N), and occurring on all continents except Antarctica, where no data were available (Figs. 1–3). Phosphorus limitation of aboveground plant production occurred on all studied continents (Figs. 1–3), although the proportion of P limitation instances differed among ecosystem types (Supplementary Table 3). Globally, 301 of the 652 experiments (46.2% of all experiments; 45.0% of experiments in natural terrestrial ecosystems; and 48.6% of experiments in croplands) revealed significant P limitation of aboveground plant production (Fig. 1). These findings provide convincing evidence that P limitation of aboveground plant production in terrestrial ecosystems is a worldwide phenomenon.

The conventional notion that P limits aboveground plant production mainly in tropical regions is based on the following patterns: relative to temperate regions, tropical regions generally have older and more weathered soils[2], higher plant N:P ratios[21,22], higher plant P use efficiencies[22,23], and lower plant and soil P concentrations[2,24]. All of these latitudinal patterns, however, can only indicate the relative magnitude rather than the actual magnitude of nutrient limitation across regions. The actual magnitude of nutrient limitation is determined most reliably by experiments in which the response of aboveground plant production to nutrient addition is quantified[3,25]. Our meta-analysis of P-fertilization field experiments shows that P significantly limits aboveground plant production across tropical, subtropical, temperate, and (sub)arctic regions, although both the magnitude of P limitation and the percentage of P limitation instances were greater in tropical and subtropical regions than in temperate and (sub)arctic regions (Fig. 3 and Supplementary Table 3).

The worldwide occurrence of P limitation of terrestrial aboveground plant production may be explained by the biochemical machinery shared by autotrophs[1,26], the great variability of plant characteristics and environmental conditions within regions[8], and the multiple pathways resulting in P limitation of aboveground plant production[4]. Researchers have proposed that the P and N demands of core biochemical machinery (mainly concerning rRNA and proteins) shared by all photoautotrophs may cause plant growth to be limited by P and N to a similar degree[1,26]. Although ecosystem properties such as soil P availability are key drivers of P limitation of aboveground plant production in terrestrial ecosystems[8,27], ecosystem properties vary greatly across sites[8]. For example, the soil P supply rate ranges from ~1 to >10000 g $m^{-2}$ $yr^{-1}$ in both tropical and temperate regions[8]. P limitation in tropical regions is often attributed to the occlusion of P in soil by chemical or physical mechanisms, the chronic loss of dissolved inorganic and organic P by leaching, and/or the exhaustion of soil primary minerals during long-term soil development[7]. However, there are also other pathways that can cause P limitation in different regions and at different timescales (from years to millions of years), including the formation of soil layers (e.g., iron pans) that physically prevent/inhibit access by roots to potentially available P, transactional limitations in which the input of P by weathering is less than the input of other resources, low-P parent material, sinks that reduce P levels, and anthropogenic increases in the supply of other resources and especially N and atmospheric $CO_2$[4]. Permafrost, for example, can isolate plants from deeper portions of the soil profile in cold regions[4,28]. Low-P parent materials explain P limitation in some temperate regions[8]. These and other pathways (e.g. the precipitation of P with Ca in arid soils) can cause P limitation in many temperate and (sub)arctic regions.

**The magnitude of P limitation**. In natural terrestrial ecosystems, P additions increased aboveground plant production over controls

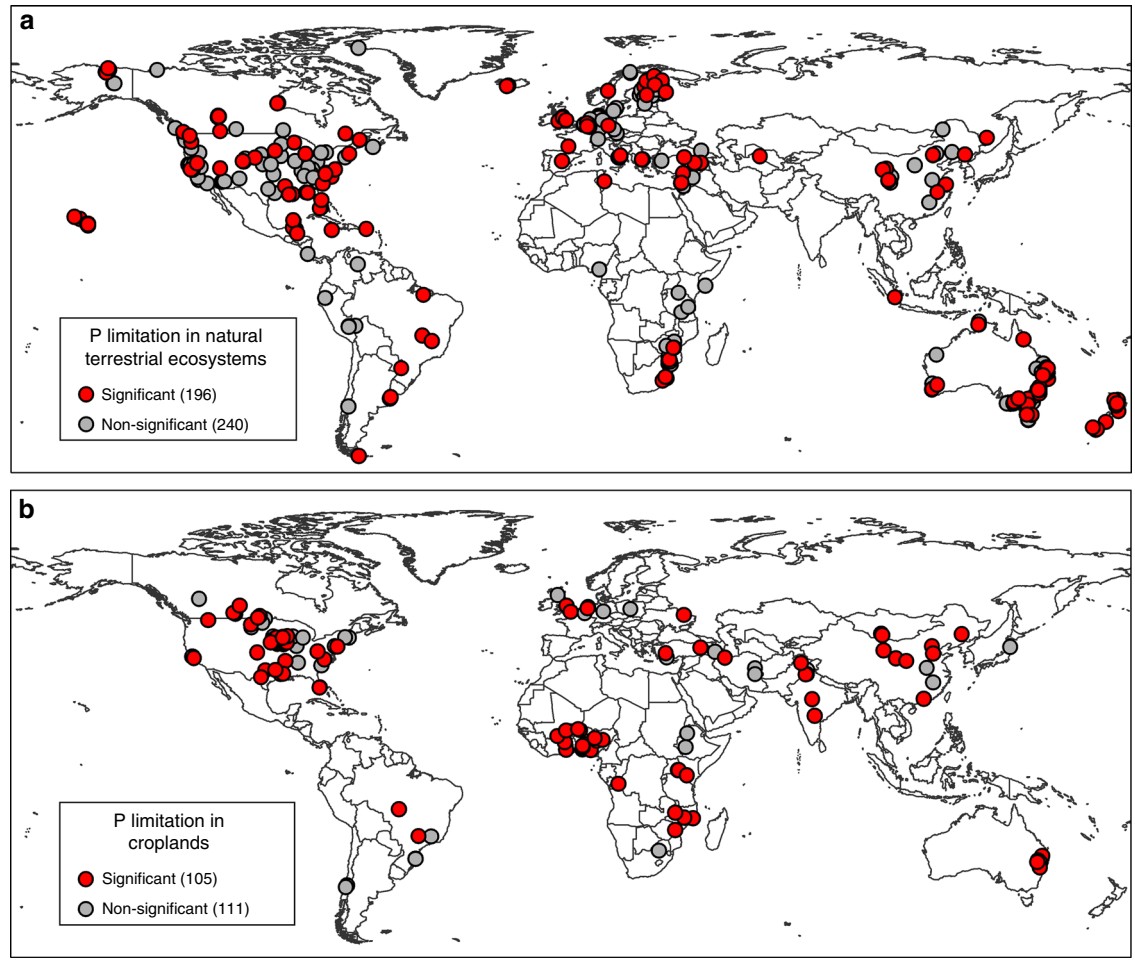

**Fig. 1 Locations of the 652 experiments in which the effect of P addition on aboveground plant production was assessed. a** Natural terrestrial ecosystems. **b** Croplands. Experiments were determined to have significant P limitation based on the Ln(Response Ratio). If the Ln(Response Ratio) was higher than a threshold value (0.23 for natural terrestrial ecosystems and 0.09 for croplands), it was considered a significant case (Z test, P < 0.05) of P limitation. Determination of the threshold values is described in the "Methods" section and is supported by the Supplementary Fig. 3. Numbers in brackets are the number of experiments in the indicated group. Source data are provided as a Source Data file.

by an average of 34.9%, with a 95% confidence interval of 30.0%–40.1% (N = 436; Table 1). The estimates were robust given our large sample size, as suggested by our sensitivity tests (not sensitive to outliers; Supplementary Fig. 4), publication bias tests (no significant publication bias; Supplementary Fig. 5a) and temporal change test (a significant but minor change in effect size with publication year; Supplementary Fig. 6a). Our average (34.9%) was about two times greater than the average reported in a recent global meta-analysis (17.7%)[15] that used the same meta-analysis method but with a much smaller sample size (N = 60). To compare our meta-analysis with three other previous meta-analyses[1,8,9], we also calculated the magnitude of P limitation in the natural terrestrial ecosystems by weighting the Ln(RR) uniformly or by weighting the RR with the inverse variance (see details in "Methods" section). The estimated averages (36.3% and 47.6%, respectively) were again higher than those reported in the previous meta-analyses (23.4%[8]–29.3%[1] and 31.7%[9], respectively, N = 50–117; Table 1). Moreover, we found that the P effect size increased with quantity of P added and with the experimental duration (Fig. 3). Therefore, the magnitude of P limitation in the natural terrestrial ecosystems was even larger after the Ln(RR) was weighted by the quantity of P added (40.5%) or experimental duration (48.4%) (Table 1).

The estimate for natural terrestrial ecosystems was lower in Elser et al.[1] than in our study (Table 1). This perhaps because up

to 41.1% (N = 44) and 43.0% (N = 46) of the 107 terrestrial P-addition experiments in the study by Elser et al.[1] were performed in Europe and North America, respectively (Supplementary Fig. 1b). The P effect size in North America (36.9%) in the current study was close to the global average, but the P effect size in Europe (21.7%) was much smaller than the global average and the averages in Australia (50.6%), Asia (40.4%), and South America (37.7%) (Fig. 3). Therefore, the global magnitude of P limitation is much larger than that based on data mainly from Europe and North America. The experiments in Augusto et al.[8] were spread quite evenly over the global land surface (Supplementary Fig. 1c). Their relatively low estimates might be partly explained by their use of a slightly different way (relative to our study) to remove pseudo-replications, i.e., the latest measurement for forests but the earliest measurement for other ecosystems. Moreover, all of the previous syntheses had a much smaller sample size (Table 1) and therefore a relatively poorer representation of global natural terrestrial ecosystems than our study (Supplementary Figs. 1, 4).

The P effect size was much smaller in croplands than in natural terrestrial ecosystems (Table 1) and was even smaller after it was adjusted with the trim-and-fill method (4.1%; Supplementary Fig. 5b). The pattern holds true for most continents and climate zones (Fig. 3), under all fertilization regimes (Fig. 3), on all types of soils (Supplementary Fig. 7), and for all meta-analytic methods

**Table 1 The magnitude of P limitation in natural terrestrial ecosystems is larger than previously estimated.**

| Method | This study | | Previous studies | |
|---|---|---|---|---|
| | Effect size [Lower *CI*, Upper *CI*] (%) | Number of experiments | Effect size [Lower *CI*, Upper *CI*] (%) | Number of experiments |
| Natural terrestrial ecosystems | | | | |
| Ln(*RR*) weighted by the inverse variance | 34.9 [30.0, 40.1] | 436 | 17.7 [11.1, 24.8][15] | 60 |
| Ln(*RR*) weighted uniformly | 36.3 [31.0, 41.9] | 436 | 29.3 [14.9, 45.4][1] | 107 |
| | | | 23.4 [16.8, 30.4][8] | 117 |
| RR weighted by the inverse variance | 47.6 [34.5, 60.8] | 436 | 31.7 [21.6, 41.8][9] | 50 |
| Ln(*RR*) weighted by P addition amount | 40.5 [27.5, 54.8] | 436 | | |
| Ln(*RR*) weighted by experimental duration | 48.4 [33.5, 64.9] | 436 | | |
| Croplands | | | | |
| Ln(*RR*) weighted by the inverse variance | 13.9 [11.1, 16.8] | 216 | | |
| Ln(*RR*) weighted uniformly | 16.3 [13.0, 19.7] | 216 | | |
| RR weighted by the inverse variance | 15.4 [7.0, 23.7] | 216 | | |
| Ln(*RR*) weighted by P addition amount | 24.7 [11.2, 39.8] | 216 | | |
| Ln(*RR*) weighted by experimental duration | 25.9 [12.0, 41.6] | 216 | | |

The magnitude of P limitation in the natural terrestrial ecosystems and also in the croplands was calculated using the five methods listed in the table.
*CI* indicates confidence interval. Ln(*RR*) indicates ln transformed response ratio.

(Table 1). The pattern can be at least partly explained by the higher availability of soil P in croplands than in natural terrestrial ecosystems (Supplementary Fig. 8). Before P-addition experiments were performed, some of the croplands likely received P fertilizers that increased soil P availability, although no pre-experiment fertilization was recorded in the source literature (see "Methods" section for the selection criteria). In croplands in France, for example, an average of 82% of soil P was estimated to have originated from former fertilization applications[29]. Our results, therefore, suggest that P limitation in croplands has been largely alleviated by historical fertilizations[30,31] and that a reduced amount of P fertilizer is needed to increase crop production in the future. It follows that to accurately predict the future fertilizer effect on crop production, models require fertilization history. Our results concerning differences between croplands and natural terrestrial ecosystems may also be related to the lower soil organic matter contents and shorter experimental durations in croplands (Fig. 3 and Supplementary Table 1).

A positive asymmetric distribution of the P effect size was observed in the croplands (Supplementary Fig. 5b). This does not necessarily indicate a publication bias (i.e., the tendency of journals to favor the publication of statistically significant results)[32]. Fertilization experiments in croplands typically have multiple nutrient (e.g., P, N, and K) treatments, and P addition is only one of the multiple treatments. Therefore, there is no apparent tendency of journals to favor publication of statistically significant P effects. Instead, the asymmetric distribution may mainly result from a true heterogeneity[32]. The growth of plants in some regions is known to be strongly limited by P supply (e.g., Australia and lowland tropical areas with strongly weathered soils)[4,7]. However, we are unaware of any large, negative P effect on plant growth at the community level, although there are rare reports of P toxicity symptoms in some plants that have adapted to low soil P availability[33] and of P-driven limitation of plant growth by N via soil microbes in some N-limited ecosystems[27]. Therefore, the detection of many more positive P effects than negative P effects is reasonable (Supplementary Fig. 5).

**Predictors of the magnitude of P limitation.** Although our results show that P limitation is a global phenomenon, the magnitude of P limitation did vary greatly among experiments, with the Ln(*RR*) ranging from −0.48 to 2.44 (Fig. 2). Climate,

fertilization regimes, soil properties, and plant properties each explained some percentage (9.1%–40.0%) of the total explained variation in P effect size in both the natural terrestrial ecosystems ($R^2 = 0.59$) and the croplands ($R^2 = 0.79$) (Fig. 4 and Supplementary Figs. 9–11. That the P effect size is regulated by multiple factors rather than by any single factor once again explains why P limitation is widespread on the globe. For example, P limitation can occur in the tropical and subtropical natural ecosystems due to the high temperatures and precipitation that drive plant P demand and to the low soil extractable P concentration that limits soil P supply[24] (Supplementary Fig. 10a, c, d). In contrast, the occurrence of P limitation in the temperate and (sub)arctic natural ecosystems may be attributed to their generally high soil organic matter contents and pH values (Supplementary Fig. 10e, h). High soil organic matter content can reduce soil P availability by occluding P in the organic forms and by enhancing microbial immobilization of P in the soil[25,27,34]. High soil pH can reduce soil P sorption capacity[35] and thus increase the use efficiency of P fertilizer by plants[25] (i.e., increase the P effect size). Moreover, both high soil organic matter content and moderate soil pH can enhance the availability of nutrients such as N, potassium, and calcium in soils[35,36], which may exaggerate the response of plant growth to P addition[9,31,37].

As our study did for P, the study of LeBauer and Treseder[6] examined plant responses to N additions alone; they found a global distribution of N limitation on terrestrial primary production. Elser et al.[1] and Harpole et al.[37], however, reported a prevalent co-limitation of terrestrial primary production by N and P. Although the results seem conflicting (i.e., globally distributed P limitation, N limitation, or N and P co-limitation), they can be reconciled by the multiple limitation hypothesis[27,37]. A prevalent co-limitation of terrestrial primary production by N and P suggests a generally balanced N and P limitation in global terrestrial ecosystems[1,37], while a globally distributed N limitation indicate widespread N limitation in terrestrial ecosystems[3,6]. Given that both are reasonable, a worldwide occurrence of P limitation in terrestrial ecosystems is expected, as observed in the present study; the absence of a worldwide occurrence of P limitation would either imply an imbalance of N and P limitation, which would counter the finding of prevalent N and P co-limitation (if widespread N limitation is true), or imply a less widespread N limitation, which would counter the finding of globally distributed N limitation (if prevalent N and P co-limitation is true). Furthermore, the widespread P limitation

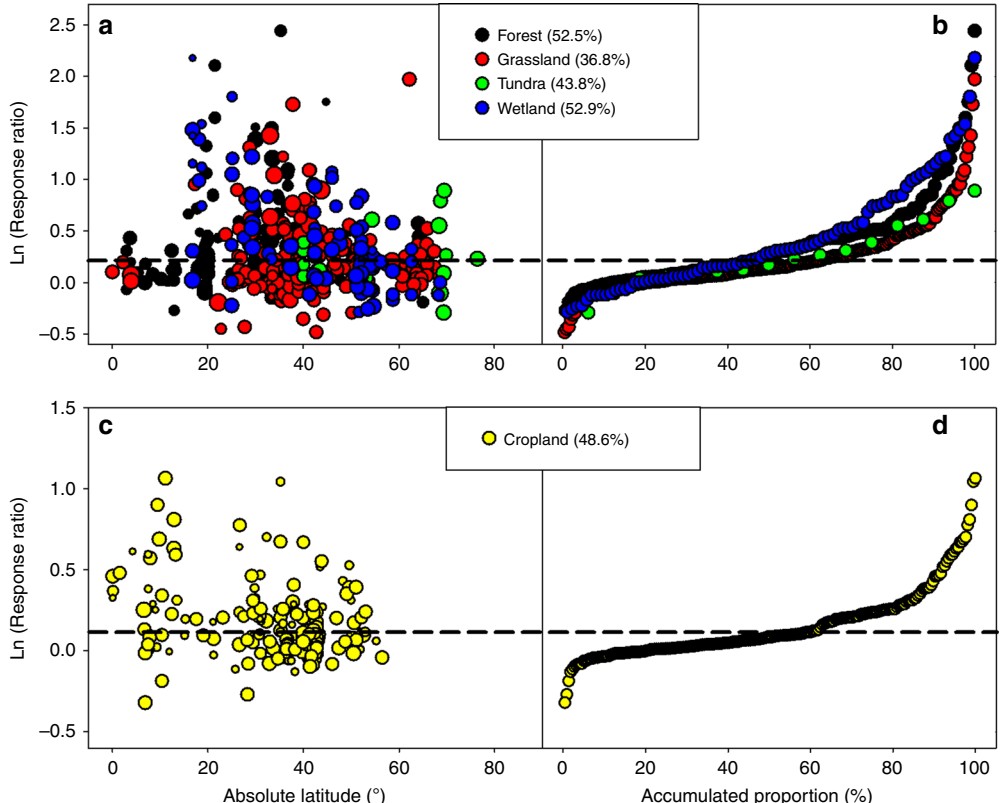

**Fig. 2 Consistent occurrence of significant P limitation in all types of ecosystems. a** Significant P limitation occurred in the natural terrestrial ecosystems at almost all latitudes. The magnitude of P limitation decreased with latitude in wetlands (meta-regression, $R^2 = 0.35$, $P < 0.05$, $N = 85$) but not in any other type of natural terrestrial ecosystem (meta-regression, $P > 0.05$). **b** Significant P limitation occurred in all types of natural terrestrial ecosystems. Values in the brackets indicate the percentage of significant P limitation cases in the total sample size in each type of natural terrestrial ecosystem. **c** Significant P limitation occurred in croplands at all explored latitudes (absolute latitude between 0.1° and 56.5°), and the magnitude of P limitation decreased with absolute latitude (meta-regression, $R^2 = 0.16$, $P < 0.05$, $N = 216$). **d** Significant P limitation was found in 48.6% of the P-fertilization experiments in the croplands. As in Fig. 1, statistical significance of P limitation was assessed based on the Ln(Response Ratio). Dashed lines in all plots (0.23 in (**a**) and (**b**), 0.09 in (**c**) and (**d**)) indicate where the magnitude of P limitation approximates a 0.05 significance level (Z test). In (**a**) and (**c**), the size of each point is proportional to the weight used for meta-regression analysis. Source data are provided as a Source Data file.

identified in this study and the widespread N limitation reported in LeBauer and Treseder[6] together imply that P limitation and N limitation are largely independent of each other[37]. This possibility is supported by another synthesis study, which reported that the effects of N supply and P supply on aboveground plant production are additive in most terrestrial ecosystems[15]. Taken together, the evidence suggests the worldwide occurrence of both P limitation and N limitation in terrestrial ecosystems[8], which supports the multiple limitation hypothesis and challenges Liebig's Law of the Minimum[27,37].

Although our dataset is much larger than those in previous syntheses, there is still a large uncertainty in our estimate of the global magnitude of P limitation in terrestrial ecosystems. There are three sources of this uncertainty. First, while ecosystems in Australia, East Asia, and West Asia are better represented in our dataset than in the previous ones, ecosystems in North Asia and the tropics are still largely underrepresented (Supplementary Fig. 1a). More experiments from North Asia may lower the global averages of P limitation, while more experiments from the tropics would likely increase the global averages. Mature mixed forests were also underrepresented (Supplementary Table 3), and the inclusion of an increased number of forests may lower the global average of the natural terrestrial ecosystems (Supplementary Fig. 7). Second, most experiments were performed for $\leq 10$ yrs and with a cumulative P addition $\leq 500\,\mathrm{kg\,ha^{-1}}$, which may not be long enough or high enough to fully stimulate the growth of

plants in natural terrestrial ecosystems (Fig. 3 and Supplementary Fig. 10). This may lead to an underestimation of the global average of P limitation in natural terrestrial ecosystems. Moreover, additional uncertainties can be introduced by the statistical analyses (Table 1), missing variances of aboveground plant production (Supplementary Table 4), and the various measures of aboveground plant production used in different experiments (Supplementary Table 5). Missing measurements of ecosystem properties such as soil extractable P concentration and pH (Supplementary Table 1) can lead to an underestimation of their relative importance in predicting the magnitude of P limitation. Third and last, given the long span of time of the datasets (Supplementary Fig. 2), the nature of nutrient limitation has likely changed over much of the land covered in this analysis, due, for example, to the changes in atmospheric N deposition[38] and to changes in fertilization practices in the croplands[30]. In spite of these uncertainties, our large dataset of P addition experiments provides a much clearer pattern of the global distribution of P limitation and a more robust estimate of the global magnitude of P limitation in terrestrial ecosystems than previous datasets.

Our findings have important implications for understanding the role of P supply in controlling aboveground plant production in terrestrial ecosystems. The results show a more widespread and much stronger limitation of aboveground plant production by P in natural terrestrial ecosystems than previously thought. The results confirm the necessity of incorporating P limitation in

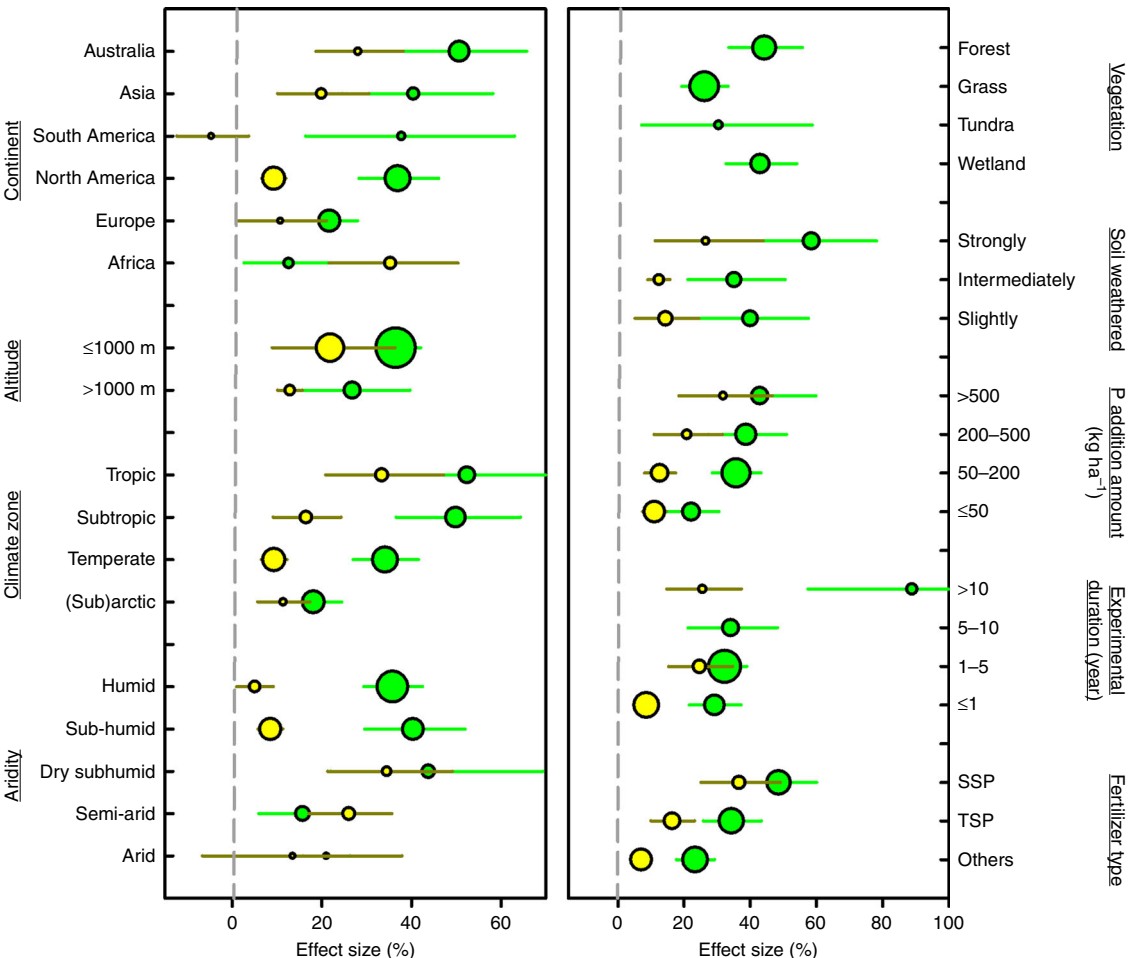

**Fig. 3 P limitation was significant in all regions and major types of ecosystems and under various fertilization regimes.** Exceptions are the non-significant P limitation in two groups of experiments in the croplands that had a small sample size ($N = 8$). Natural terrestrial ecosystems are shown in green (total $N = 436$), and croplands are shown in yellow (total $N = 216$). Values represent effect sizes ± 95% confidence intervals. The size of each point is proportional to the sample size (sample sizes are listed in Supplementary Table 3). The dashed lines indicate the no-fertilization effect. SSP is single superphosphate, and TSP is triple superphosphate. Source data are provided as a Source Data file.

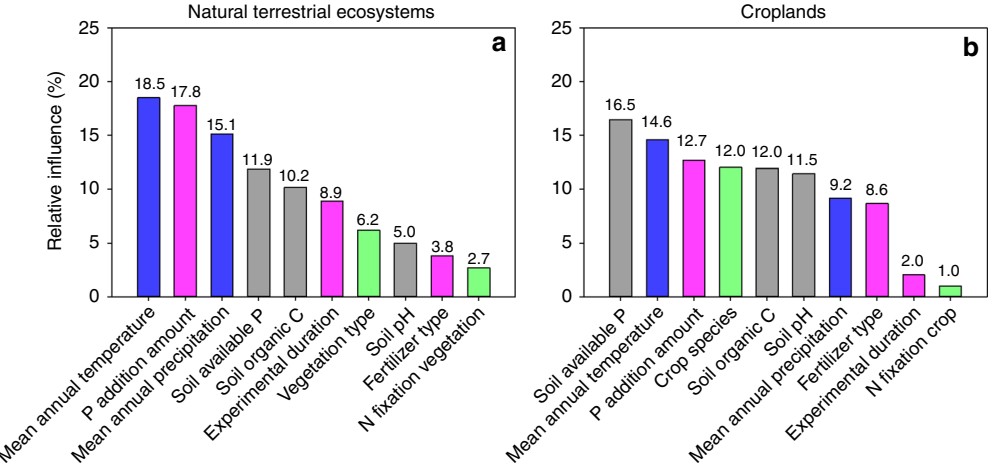

**Fig. 4 Relative influence of climate, fertilization regimes, and ecosystem properties on the magnitude of P limitation. a** Natural terrestrial ecosystems. **b** Croplands. The number above each bar indicates the percentage of the total explained variation accounted for by the variable. Fertilization regimes are in pink, climate factors are in blue, soil properties are in gray, and vegetation properties are in green. Source data are provided as a Source Data file.

earth system models[13,14]. The results also show a much smaller P fertilizer effect in croplands than in natural terrestrial ecosystems, which suggests that P limitation in croplands has been generally alleviated by historical fertilizations. Finally, the co-regulation of plant response to altered P supply by climate, ecosystem properties, and fertilization regimes highlights the importance of taking a systems approach to study how nutrient supply affects aboveground plant production.

## Methods

**Data collection**. With the aim of constructing a comprehensive database of the experimentally determined effects of P additions on aboveground plant production in global terrestrial ecosystems, we collected as many experiments that fulfilled our criteria (described below) as possible. Relevant studies were identified by searching *ISI Web of Knowledge*, *Google Scholar*, and *China Knowledge Resource Integrated Database* using combinations of keywords such as "phosph* addition", "phosph* fertili*", "phosph* enrich*", "aboveground biomass", "primary product*", "crop yield", and "grain yield". Our survey also included studies summarized in previously published syntheses and the subsequent relevant studies citing those syntheses. A PRISMA flow diagram (Supplementary Fig. 12) shows the procedure we used for the selection of studies.

To be included in our database, published experiments were required to satisfy the following criteria: (1) the P-addition experiment was conducted in the field and included P-addition and control treatments within the same ecosystem under the same environmental conditions, and also included measures of aboveground plant production in both P-addition and control treatments; (2) no fertilization was recorded in the control treatment either before the start of the experiment or during the experiment; (3) the P treatment received a P fertilizer that did not contain N so as to avoid the effect of N; as a result experiments with application of ammonium phosphate, manure or other fertilizers were excluded.

To be considered an experiment in our analysis, a reported experiment had to be temporally and spatially distinct and had to have internally consistent controls. Multiple experiments could be reported by one publication; for instance, the application of the same experimental treatments was considered to represent multiple experiments if the treatments were applied at several sites with different vegetation types. When multiple measures were reported over time at a single experimental site, we used the latest measure. When multiple levels of P fertilizer treatments were reported, we used the measure with the highest amount of P addition. Choosing the latest measure and the highest P addition amount increased the likelihood that P additions fulfill plant demand and overcome the sorption of P fertilizer by soils and soil microbial competition for P fertilizer[25,27]. When multiple forms of P fertilizers were tested, we chose the treatment of single superphosphate or triple superphosphate if available.

We included only experiments that reported the response of community-level aboveground plant production to P additions. Single-species responses were not included unless drawn from a mono-dominant community. If several species from a community were individually assayed, an average across all species was used. Experiments in forest or savanna ecosystems that only reported the response of understory or herbaceous response to P additions were not included. Experiments with only stand biomass responses were excluded unless the stand biomass data could be used to calculate aboveground plant biomass production.

Ecosystems were classified as forest, grassland, tundra, wetland, or cropland; natural forests, plantations, shrublands, and savannas were all classified as forest. In forest ecosystems, beside aboveground plant biomass production ($N = 33$), we also accepted proxy variables that are known to be correlated with aboveground plant biomass production, such as litterfall production (1) and the rate of increase in tree diameter (34), stem volume (25), basal area (25), or height (16) (Supplementary Table 5). We showed that the weighted Ln($RR$) did not differ significantly among the various variables used (Supplementary Table 5). In croplands, beside aboveground plant biomass production ($N = 85$), we also accepted marketable yield (131), because we found that marketable yield responded similarly to P additions as aboveground plant production in the croplands based on studies that reported both measures (Supplementary Fig. 13). In tundras, beside aboveground biomass production ($N = 10$), we also included leaf mass per tiller (4), tiller biomass (1), and plot level NDVI (1) (Supplementary Table 5). In wetlands, beside aboveground biomass production ($N = 72$), we also included height increase (5), leaf area index (3), the production of whole plants (3), and chamber based gross primary production (2) (Supplementary Table 5).

In total, we collected data from 652 P addition experiments reported in 285 published papers, including 436 experiments from natural terrestrial ecosystems (including forests, grasslands, tundras, and wetlands) and 216 experiments from croplands (see experimental locations in Supplementary Fig. 1a). Beside aboveground plant production measures, our database also included site characteristics and fertilization regimes, which were used to explain the variation in Ln($RR$). Site characteristics included site location (latitude and longitude), climate variables (MAT and MAP), topographical conditions (altitude and slope), plant characteristics (vegetation type, and symbiotic N fixation), soil type, soil physiochemical properties before the experiments began or from the control

treatments (concentrations of available P, organic C, and total N; pH in water; and particle size), and parent material type. For each experiment in forest ecosystems, forest composition (i.e., pure or mixed forest) and the average forest age during the experiment were also recorded.

**Data preparation**. In cases where the referenced studies did not report the latitude or longitude of the P-addition experiment (52% of the studies did not report both latitude and longitude), the approximate latitude and longitude were derived by geocoding the name of the location in Google Earth 7.0 (the free version). In cases where the referenced studies did not report MAT (76%), MAP (54%), or altitude (65%), the values were derived from WorldClim[39] using site geographic location (i.e. latitude and longitude). The aridity index (AI) of each site was obtained from CGIAR-CSI using data from WorldClim[40]; the AI value decreases as aridity increases.

Soil type was classified according to the U.S. Department of Agriculture soil classification system[41]. Soils were grouped based their degree of weathering according to previous studies[42,43]: Andisols, Histosols, Entisols, and Inceptisols were considered to be slightly weathered soils; Aridsols, Vertisols, Mollisols, and Alfisols were considered to be intermediately weathered soils; and Spodosols, Ultisols, and Oxisols were considered to be strongly weathered soils. Parent material types were grouped into four geological classes according to a previous study[8]: acid, intermediate, mafic, and calcareous rocks.

For comparison of P effect sizes among regions, experiments in the database were grouped in four different ways. First, experiments were grouped according to their continental locations: Australia, Asia, Africa, Europe, North America, and South America. Second, experiments were grouped based on absolute latitude into four latitude belts or regions: tropic (23.4 ºS–23.4 ºN), subtropic (23.4–35 ºS or ºN), temperate (35–50 ºS or ºN), and (sub)arctic (>50 ºS or ºN). Third, experiments were grouped according to altitude into low-altitude experiments (< 1000 m a.s.l.) and high-altitude experiments (≥1000 m a.s.l.). Finally, experiments were divided based on site aridity level into five groups: arid (AI ≤ 0.20), semi-arid (0.20 < AI ≤ 0.50), dry subhumid (0.50 < AI ≤ 0.65), sub-humid (0.65 < AI ≤ 1.0), and humid (1.0 < AI). The complete dataset is available at Figshare[44].

**Phosphorus limitation threshold**. One major objective of our study was to map the global distribution of experiments in which P significantly limited aboveground plant production. To do this, we had to define a threshold value that separated experiments that did or did not find significant P limitation. We estimated the threshold value separately for the natural terrestrial ecosystems and the croplands, using a method described in a recent study[8]. In general, we first collected the reported statistical responses of aboveground plant production to P additions from the source references. We then investigated the distribution of the Ln($RR$) values. Finally, we identified the threshold value of the Ln($RR$) that optimizes the distinction between statistically significant positive P effects and statistically non-significant P effects. Of the 128 experiments in the natural terrestrial ecosystems that reported a significant P limitation, 84% had an Ln($RR$) value ≥ 0.23 (Supplementary Fig. 3a). Similarly, of the 162 experiments in the natural terrestrial ecosystems that reported a non-significant P effect, 85% had an Ln($RR$) < 0.23 (Supplementary Fig. 3b). When the two groups were combined, the maximum percentage (84%) of correct classification (i.e., a significant positive effect was classified as a significant case and a non-significant effect was classified as a non-significant case) was obtained with an Ln($RR$) value of 0.23 (Supplementary Fig. 3c). Therefore, 0.23 was used as the threshold Ln($RR$) value to distinguish significant from non-significant P limitation in natural terrestrial ecosystems. This threshold value is close to the one used in a previous study (Ln($RR$) of 0.20)[8]. A similar approach was applied to the P-addition experiments in the croplands, such that 0.09 was used as the threshold Ln($RR$) value for croplands (Supplementary Fig. 3d–f).

**Meta-analysis**. We quantified the magnitude of P limitation at the global scale and in various groups of ecosystems by weighting the Ln($RR$) with the inverse variance and a random-effect model[16–19]. To do this, we extracted means, standard deviations (SDs), and sample sizes ($n$) from the published studies. If standard error (SE) rather than SD was reported, SD was calculated:

$$SD = SE\sqrt{n} \tag{1}$$

If neither SD nor SE was reported, we approximated the missing SD by multiplying the reported mean by the average coefficient of variance of our complete dataset. If sample size was not reported, we assigned sample sizes as the median sample size of our complete dataset. We approximated the SDs and the sample sizes separately for the natural terrestrial ecosystems and the croplands and also separately for the control and the P-addition treatments (see details in Supplementary Table 4).

The Ln($RR$) of an experiment was calculated as follows:

$$\text{Ln}(RR) = \text{Ln}\frac{\overline{X_t}}{\overline{X_c}} = \text{Ln}(\bar{X}_t) - \text{Ln}(\bar{X}_c) \tag{2}$$

where $\bar{X}_t$ and $\bar{X}_c$ are mean aboveground plant production in the P treatment and control, respectively.

The weighted mean response ratio ($Ln(RR_+)$) of a group of ecosys was follows:

$$Ln\left(RR_+\right) = \frac{\sum_{i=1}^{m} w_i^* \times Ln(RR_i)}{\sum_{i=1}^{m} w_i^*} \qquad (3)$$

where $m$ is the number of experiments in the group (e.g., a region), and $w_i^*$. is the weighting factor of the $i$th experiment in the group. The $w_i^*$. was calculated as follows:

$$w_i^* = \frac{1}{v_i^*} \qquad (4)$$

where $v_i^*$. is the variance of study ($i$) in the group. The $v_i^*$. was calculated as follows:

$$v_i^* = v_i + T^2 \qquad (5)$$

where $v_i$ is the within-study variance of study ($i$), and $T^2$ is the between-studies variance. The $v_i$ was calculated as follows:

$$v_i = \frac{S_t^2}{n_t \overline{X}_t^2} + \frac{S_c^2}{n_c \overline{X}_c^2} \qquad (6)$$

where $n_t$ and $n_c$ are the sample sizes for the P treatment and the control groups, respectively, and $S_t$ and $S_c$ are the standard deviations for the P treatment and the control groups, respectively, of study ($i$). The calculation of $T^2$ can be seen in Borenstein et al.[45].

The standard error of the $Ln(RR_+)$ was calculated as:

$$s\left(Ln\left(RR_+\right)\right) = \sqrt{\frac{1}{\sum_{i=1}^{m} w_i^*}} \qquad (7)$$

The 95% confidence interval ($CI$) for the $Ln(RR_+)$ was calculated as follows:

$$95\%CI = Ln\left(RR_+\right) \pm 1.96 s\left(Ln\left(RR_+\right)\right) \qquad (8)$$

If the 95% $CI$ did not overlap with zero, the overall P addition effect in the group of experiments was considered significant. The percentage change in aboveground plant production induced by P addition (i.e., the effect size) in a group of ecosystems was measured as follows:

$$\text{Effect size}(\%) = (\exp\left(Ln\left(RR_+\right)\right)1)100\% \qquad (9)$$

The meta-analyses were performed using "$meta$" package in $R$ version 3.3.1[46].

To compare our analyses with the previous meta-analyses[1,8,9], we also calculated the global magnitude of P limitation using two other methods:

(1) $Ln(RR)$ weighted uniformly[1,8], where effect size only depends on the means of control and P treatment groups.
(2) $RR$ weighted by the inverse variance[9], where the $RR$ rather than the $Ln(RR)$ is weighted by the inverse variance.

We also calculated the global magnitude of P limitation by considering the effects of the quantity of P added and experimental duration:

(3) $Ln(RR)$ weighted by the quantity of P added. The weight of each experiment was calculated as follows:

$$w_P = \frac{P}{\sum_{i=1}^{M} P_i} \qquad (10)$$

where $P$ is the cumulative quantity of P added during the experiment (kg ha$^{-1}$), and $M$ is the number of experiments in the natural terrestrial ecosystems (436) or the croplands (216).

(4) $Ln(RR)$ weighted by experimental duration. The weight of each experiment was calculated as follows:

$$w_D = \frac{D}{\sum_{i=1}^{M} D_i} \qquad (11)$$

where $D$ is the duration of the experiment (yr).

We checked the quality of our meta-analysis according to the checklist of Koricheva and Gurevitch[19]. Our meta-analysis fulfilled all of the quality criteria for a meta-analysis in plant ecology (Supplementary Table 6). We performed two sensitivity tests (Supplementary Fig. 4). One was the leave-one-out meta-analysis using the "$metainf$" function. The other was the cumulative meta-analysis using a random-effect model in the "$metacum$" function based on inverse-variance weighted $Ln(RR)$. The cumulative meta-analysis was repeated 1000 times with random orders of experiments. We also ran a cumulative meta-analysis using the "$metacum$" function based on the uniformly weighted $Ln(RR)$. We created funnel plots to detect possible publication bias using the "$funnel$" function. The possible publication bias was statistically tested using the "$metabias$" function. If a publication bias was suggested, we further adjusted the P effect size by the trim-and-fill method using the "$trimfill$" function. Finally, we explored the relationships between the $Ln(RR)$ and the absolute value of latitude and publication year using the "$metareg$" function. All the above functions were from the "$meta$" package in $R$ version 3.3.1[46].

**Boosted regression tree analysis.** Boosted regression tree (BRT) analyses were conducted to quantify the relative importance of climate, ecosystem properties, and fertilization regimes in predicting the $Ln(RR)$ in the natural terrestrial ecosystems and in the croplands. Before BRT analyses, variable selections were made to avoid high correlations among predictors. Specifically, (1) Soil organic C concentration was included as an indicator of soil organic matter content, while soil total N concentration was not included in the BRT models due to its high correlation with soil organic C concentration (Natural terrestrial ecosystems: $r = 0.85$, $P < 0.001$, $N = 92$; Croplands: $r = 0.95$, $P < 0.001$, $N = 54$). (2) Soil available P concentration was included as an indicator of soil P availability, while soil total P concentration was not included. Ecosystem properties such as soil particle size and parent material type were not included in our BRT analyses due to the very large proportions of missing data (Supplementary Table 1), including which can bias the estimate of their relative importance.

Parameter values used for the BRT analyses generally followed the recommendation of a previous study[20], i.e. bag fraction as 0.75, the number of cross validation as 10, and tree complexity as 2. Learning rate was set at a small value (i.e., 0.005) to include a large number (>1000) of regression trees in the models. Because $Ln(RR)$ is a continuous numerical variable, a Gaussian distribution of errors was used. The relative importance of each predictor represented a percentage of the total variation explained by the models. The BRT analyses were performed with the "$gbm$" package version 2.1.5[47] plus the custom code of another study[20] in R version 3.3.1. For evaluation of the spatial structure of the BRT residuals, the global Moran's $I$ statistic was applied to determine the significance using the "$spdep$" package version 0.7.7[48].

**Reporting summary.** Further information on research design is available in the Nature Research Reporting Summary linked to this article.

## Data availability
All data used in this study are available at Figshare (https://doi.org/10.6084/m9.figshare.8969963). The source data underlying all Figures except Supplementary Figs. 5, 10, and 11 are also available on the above web page. Supplementary Figs. 5, 10, and 11 are directly created using $R$ functions, as described in Methods.

## Code availability
The code used in this study is available at Figshare (https://doi.org/10.6084/m9.figshare.8969963).

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

## Acknowledgements

We thank all of the authors whose data were included in our study and Marijke Heenan and Bruce Jaffee for improving the language. This work was supported by the National Natural Science Foundation of China (31870464; 41973076; 41471443; 41401326), the US Department of Energy (DE-SC00114085), National Science Foundation Grant DEB 1655499, US Department of Energy, Terrestrial Ecosystem Sciences Grant DE-SC0006982, and the subcontracts 4000158404 and 4000161830 from Oak Ridge National Laboratory (ORNL) to the Northern Arizona University.

## Author contributions

E.H., Y.L. and D.W. designed the study. E.H. collected the data. E.H. and Y.L. analyzed the data. E.H., Y.L., Y.K., C.C., X.Lu, L.J., X.Luo and D.W. contributed significantly to the writing of the manuscript.

## Competing interests

The authors declare no conflict of interest.
