## [Peer Review File · Nature Communications]

Reviewers' comments:

Reviewer #1 (Remarks to the Author):

The submitted manuscript reports a global study of the quantitative importance of phosphorus (P) limitation on plant growth over all kinds of terrestrial ecosystems. Although several publications already gave such a global perspective (refs. 1, 8, 23), the present manuscript is welcomed because the reference 1 needs an update, and because the references 8 and 23 focused on nitrogen (N)-P interaction (implying a smaller dataset as all "P-alone" experiments could not be taken into account). In addition, as opposed to former publications, this study also includes the croplands (and not only the "natural" ecosystems, such as forests, wetlands, and so on). However, this point is also a weakness of the study (see below).

The authors did a good job in collecting many original data for their meta-analysis. In meta-analyses, it is almost impossible to claim for data exhaustiveness. But, in this case, the amount of collected data is excellent.

The manuscript is well-written, easy to read, and can interest a large audience in the scientific community.

As a whole, this study has the potential to be published in Nature Communications and I would like to already support it. But, in its present state, I do not recommend its acceptance. Indeed, according to me, the results are statistically flawed and some of them are biased. There are three major problems in the data handling that should be fixed, and I consider that these three problems should be addressed before publication.

#1 - The use of the dataset is flawed due to pseudo-replications

The authors put forth that they collected 1983 P-experiments. This is not correct: they collected 1983 P-treatments. In many cases, there were several P-treatments in the same P-experiment. Let's have a look at the dataset (by the way, providing the dataset to reviewers is a very good practice):

Zhou et al., 2017 (code=1-3), Zhang et al., 2017 (code=4-8), Yu et al., 2017 (code=9), and Wang et al., 2017a (code=10-14), etc. It means that these four experiments (and they do be four experiments: see longitude and latitude coordinates) have respectively 3, 4, 1, and 5 P-treatments. The 5 treatments of the Wang et al's reference have exactly the same values for all the explanatory variables (climate, soil, atmospheric deposition, etc.), except the P application dose. Consequently, the 5 values of response ratio are absolutely not independent. In meta-analyses, all values should be independent, otherwise there are pseudo-replicates, which is a big and recurrent issue in science (Hurlbert, 1984). Jessica Gurevitch (who is recognized as a leader in meta-analyses in ecology (Gurevitch et al., 2018) wrote in 1999:

"Non-independence of effect-size estimates is also a problem that frequently arises in meta-analysis. The best ways to deal with it [...] remain open to debate and exploration. There are two fundamentally different kinds of non-independence, one associated with each source of variation in meta-analytic data. Both types of non-independence can lead to underestimates of the standard error of the mean effect and therefore liberal evaluations of the statistical significance of effects. If several different measurements are made on each replicate in a study (e.g., measures at several points in time or of slightly different outcomes) and different effect sizes are computed for each, the different effect sizes may be correlated because the data on which they are based are correlated. This type of dependence arises through the correlations among the within-study sampling errors and can be eliminated by using only one effect-size estimate for each set of replicates, although this approach may involve discarding at least some of the potentially relevant data provided in a study. One alternative is to conduct a different meta-analysis for each kind of effect measure."

This is clearly the case in the present study. The authors should fix this. The way to do so is clearly explained by Gurevitch & Hedges: if all P-treatments of a P-experiment have the same explanatory values, only one mean value should be retained for statistical analyses. In the example of Wang et al (code=10-14), there are 5 fold too much response values. It is only when testing for the P application dose that the authors are allowed to keep the 5 values (from the 5 P-treatments) because, in this case, the explanatory values differ from one treatment to another (the "dose" factor").

The authors might argue that a lot of meta-analyses (published in excellent journals) did not remove all those pseudo-replicates. In such a case, I would say that (1) following the wrong way

because it is the common way is not a good reason (Gurevitch & Hedges (1999): "Non-independence of effect-size estimates is also a problem that [...] arises in meta-analysis. [...]"), and (2) *Nature Communications* is a journal with a very large audience and it deserves an irreproachable methodology.

Gurevitch, J., & Hedges, L. V. (1999). Statistical issues in ecological meta-analyses. *Ecology*, 80(4), 1142-1149.

Gurevitch, J., Koricheva, J., Nakagawa, S., & Stewart, G. (2018). Meta-analysis and the science of research synthesis. *Nature*, 555(7695), 175.

Hurlbert, S.H., 1984. Pseudoreplication and the Design of Ecological Field Experiments. *Ecological Monographs*, 54(2), pp.187-211.

#2 - The results of the Boost Regression Tree (BRT) are partly biased due to lack of data. The fact that the BRT method has a high tolerance to missing values in predictors (line 262) does not mean that this machine learning method is able to simulate the results without any information. In the Table S1, it is clear that most factors have a high percentage of non-missing values (except slope), which is OK. But, in the Table S2, it is also clear that the "parent material" factor has mainly missing values. Looking at the dataset, the proportion of missing values is 88% for the parent material. In such a case, this factor should be removed from the BRT analysis: one cannot determine if the small influence of parent material (Figure 4) is due to a real low importance of this factor or just a lack of data.

#3 - Comparing croplands to natural ecosystems is interesting, but is incorrect.

The croplands respond differently from natural ecosystems to P application (see Figure S8). Moreover, it is clear that the NKM class has a huge difference of response as compared with the other fertilization treatments (see Figure 3b). This enormous difference (one order of magnitude) could be explained by the supply of all nutrients (including micro-nutrients), alleviating all nutritional limitations (whereas, it is not the case for the other fertilization treatments). By the way, based on the dataset, I estimated that 92% of the NKM values are associated to croplands. The two following procedures should be done:

- The croplands should be analyzed separately from the other ecosystems. This is important when discussing the global importance of P-limitation (above all, in comparison with former studies; lines 110-113).

- The NKM treatment should be discarded because it is obviously different from the other treatments (Figures 3b, S8, and S9; see also lines 174-176 in SI).

#Comments:

Figure 1: there are not 1983 dots-experiments!

Lines 110-113: this comparison is biased (see above)

Lines 116-117 and 119-122: incorrect. This applies only to Elser study. More recent studies (refs. 8-9) are, in practice, global.

Line 135 and Figure 4: the factor "parent material" should be discarded from the analysis because there are not enough data to calculate a reliable estimate (see above).

Methods: I suggest the authors to quantify the quality of their meta-analysis using the scales of Koricheva (Koricheva & Gurevitch (2014). *Uses and misuses of meta-analysis in plant ecology*. *Journal of Ecology*, 102(4), 828-844.) or Philibert (Philibert et al. (2012). *Assessment of the quality of meta-analysis in agronomy*. *Agriculture, Ecosystems & Environment*, 148, 72-82.).

Lines 69-70 (SI): replace "latitude" by "altitude".

Line 167 (SI): replace "altitude" by "latitude".

Lines 183-184: this comparison is flawed (see above the first main issue).

Figure S1: difficult to distinguish the croplands from the tundra. Please change the colors.

Figure S3d: this comparison is potentially biased because Eucalyptus species are generally found in tropical and subtropical areas while Pinus species are more frequent in boreal and temperate region than in tropical-subtropical regions.

Figure S9a: difficult to distinguish the class "slightly" from the class "Intermediately". Please change the colors.

In addition, replace "lightly" by "slightly".

Reviewer #2 (Remarks to the Author):

This is a well written and interesting manuscript. However, I do not think that the assertions made about the novel, insightful nature of the work are fully justified. The role of phosphorus limitations has been explored in many previous publications particularly in agricultural management and agronomy. Furthermore, I was not convinced that combining agricultural and 'natural' ecosystems was relevant or technically sounds. The removal of biomass (and phosphorus) with crop harvest leads to entirely different biogeochemical cycling pathways. Therefore, I do not recommend this manuscript for publication in this journal.

Reviewer #3 (Remarks to the Author):

The authors assemble data on plant growth responses to P fertilization and perform a meta-analysis to assess the extent of P limitation worldwide. Understanding this is critical given the anticipated role of nutrients in mediating plant responses to environmental change, especially increasing atmospheric carbon dioxide concentrations globally. In that sense, this research addresses an important topic. However, I think there are a number of shortcomings with the current version of the paper that should be addressed before it is published. I would argue that at least in its current form, the paper should not be considered for publication in Nature Communications. The authors can find my comments below.

The authors should be cautious with terminology. For example, the paper argues that this analysis includes 1983 "fertilization experiments" in the abstract and elsewhere. I can't tell for sure from the data set, but the number of independent experiments seems to be far fewer than that. To me, it would seem more appropriate to actually analyze different experiments rather than treating individual results as separate, independent data.

There is a great deal of confusion (and in some cases contradictory information) from the literature about the nature of nutrient limitation in different sites because some studies only look at the responses to single nutrient addition. For example, like this study did for P, Lebauer and Treseder (2008) only looked at plant responses to N, even when response data may have been available for both nutrients and clearly show that both are important. It has been clearly established that in many cases, co-limitation (by N and P) is the rule (Elser et al. 2007; Harpole et al. 2011). I think the authors should be both cognizant of that, and explicitly address that issue in the manuscript. Without combing through the data from all of these papers, I would suspect there are sites where plants respond to both N and P, but suggesting that those sites are (only) limited by a single nutrient adds confusion to our understanding about what the nature and extent of limitation really is. I think this needs to be addressed in the paper.

Combining the responses of agricultural and natural systems may be providing a misleading picture of nutrient limitation globally. Ag systems are often single species systems (and well-watered), whereas natural systems have much more diversity and where climate conditions may attenuate nutrient addition responses. Clearly the responses to nutrient additions would be expected to be greater in a monotypic, watered system, and those responses may be biasing the

"average" responses reported in the paper. I would suggest that the authors include an analysis of both natural and ag systems separately to examine this potential bias. Short of that, I worry that this paper may only add more confusion to the issue.

The authors should more clearly articulate how "plant production" was determined, especially in the natural systems. The database includes "mean shoot production," but how was this determined for forests? Having worked in forests my whole career, I know that it is at least as common for studies to assess just litterfall. How was this addressed here? As is, there is simply far too little detail in the methods for the reader to assess what the results mean and how they were obtained.

Line 141. Basal applications of N, K, and micronutrients? This is a huge issue I think that needs to be discussed more. how many of the studies fall into this category? I would consider removing these from the analysis, as previous additions of other potentially limiting nutrients could simply be "baiting" a P response via stoichiometric imbalances.

Line 171: Yet, the message here is that the world is P limited, and that does not bode well for careful P management without some careful caveats. Given that, I think some of the analyses I suggest above are critical to bolster the results, and especially the conclusions.

References:

- LeBauer, D. S., & Treseder, K. K. (2008). Nitrogen limitation of net primary productivity in terrestrial ecosystems is globally distributed. *Ecology*, 89(2), 371–379.
- Elser, J. J., Bracken, M. E. S., Cleland, E. E., Gruner, D. S., Harpole, W. S., Hillebrand, H., ... Smith, J. E. (2007). Global analysis of nitrogen and phosphorus limitation of primary producers in freshwater, marine and terrestrial ecosystems. *Ecology Letters*, 10(12), 1135–1142.
- Harpole, W. S., Ngai, J. T., Cleland, E. E., Seabloom, E. W., Borer, E. T., Bracken, M. E. S., ... Smith, J. E. (2011). Nutrient co-limitation of primary producer communities. *Ecology Letters*, 14(9), 852–862.

Responses to Reviewers

(Responses and changes are shown in dark blue color)

Response to Reviewer 1

Comment 1: The submitted manuscript reports a global study of the quantitative importance of phosphorus (P) limitation on plant growth over all kinds of terrestrial ecosystems. Although several publications already gave such a global perspective (refs. 1, 8, 23), the present manuscript is welcomed because the reference 1 needs an update, and because the references 8 and 23 focused on nitrogen (N)-P interaction (implying a smaller dataset as all “P-alone” experiments could not be taken into account). In addition, as opposed to former publications, this study also includes the croplands (and not only the “natural” ecosystems, such as forests, wetlands, and so on). However, this point is also a weakness of the study (see below).

The authors did a good job in collecting many original data for their meta-analysis. In meta-analyses, it is almost impossible to claim for data exhaustiveness. But, in this case, the amount of collected data is excellent.

The manuscript is well-written, easy to read, and can interest a large audience in the scientific community.

As a whole, this study has the potential to be published in Nature Communications and I would like to already support it. But, in its present state, I do not recommend its acceptance. Indeed, according to me, the results are statistically flawed and some of them are biased. There are three major problems in the data handling that should be fixed, and I consider that these three problems should be addressed before publication.

Response: Many thanks for the generally positive comments. We totally agree with the comments on the relationship between our study and the previous syntheses. We also greatly appreciate the three major comments/suggestions on our statistical analyses, which have been addressed carefully in the revised manuscript, as described in detail in the following.

Comment 2: #1 - The use of the dataset is flawed due to pseudo-replications

The authors put forth that they collected 1983 P-experiments. This is not correct: they collected

1983 P-treatments. In many cases, there were several P-treatments in the same P-experiment. Let's have a look at the dataset (by the way, providing the dataset to reviewers is a very good practice):

Zhou et al., 2017 (code=1-3), Zhang et al., 2017 (code=4-8), Yu et al., 2017 (code=9), and Wang et al., 2017a (code=10-14), etc. It means that these four experiments (and they do be four experiments: see longitude and latitude coordinates) have respectively 3, 4, 1, and 5 P-treatments. The 5 treatments of the Wang et al's reference have exactly the same values for all the explanatory variables (climate, soil, atmospheric deposition, etc.), except the P application dose. Consequently, the 5 values of response ratio are absolutely not independent. In meta-analyses, all values should be independent, otherwise there are pseudo-replicates, which is a big and recurrent issue in science (Hurlbert, 1984). Jessica Gurevitch (who is recognized as a leader in meta-analyses in ecology (Gurevitch et al., 2018) wrote in 1999:

“Non-independence of effect-size estimates is also a problem that frequently arises in meta-analysis. The best ways to deal with it [...] remain open to debate and exploration. There are two fundamentally different kinds of non-independence, one associated with each source of variation in meta-analytic data. Both types of non-independence can lead to underestimates of the standard error of the mean effect and therefore liberal evaluations of the statistical significance of effects. If several different measurements are made on each replicate in a study (e.g., measures at several points in time or of slightly different outcomes) and different effect sizes are computed for each, the different effect sizes may be correlated because the data on which they are based are correlated. This type of dependence arises through the correlations among the within-study sampling errors and can be eliminated by using only one effect-size estimate for each set of replicates, although this approach may involve discarding at least some of the potentially relevant data provided in a study. One alternative is to conduct a different meta-analysis for each kind of effect measure.”

This is clearly the case in the present study. The authors should fix this. The way to do so is clearly explained by Gurevitch & Hedges: if all P-treatments of a P-experiment have the same explanatory values, only one mean value should be retained for statistical analyses. In the example of Wang et al (code=10-14), there are 5 fold too much response values. It is only when testing for the P application dose that the authors are allowed to keep the 5 values (from the 5 P-treatments) because, in this case, the explanatory values differ from one treatment to another (the

“dose” factor”).

The authors might argue that a lot of meta-analyses (published in excellent journals) did not remove all those pseudo-replicates. In such a case, I would say that (1) following the wrong way because it is the common way is not a good reason (Gurevitch & Hedges (1999): “Non-independence of effect-size estimates is also a problem that [...] arises in meta-analysis. [...]”), and (2) Nature Communications is a journal with a very large audience and it deserves an irreproachable methodology.

Gurevitch, J., & Hedges, L. V. (1999). Statistical issues in ecological meta-analyses. *Ecology*, 80(4), 1142-1149.

Gurevitch, J., Koricheva, J., Nakagawa, S., & Stewart, G. (2018). Meta-analysis and the science of research synthesis. *Nature*, 555(7695), 175.

Hurlbert, S.H., 1984. Pseudoreplication and the Design of Ecological Field Experiments. *Ecological Monographs*, 54(2), pp.187–211.

Response: Thanks a lot for the detailed comments on the pseudo-replicate issue and the specific suggestions to solve the problem. As suggested, we recalculated the responses by selecting only a single measure from each experiment and made corresponding changes in the revision as described below.

In the revised manuscript we selected only one measurement of aboveground production from each experiment, namely the latest measure from the treatment with the highest amount of P addition. As justified for the meta-analysis of N effect in Lebauer and Treseder (2008), choosing the latest measure and the highest amount of P addition in our study increased the likelihood that P additions fulfill plant demand and overcome the sorption of P fertilizer by soils and soil microbial competition for P fertilizer. This proved true, as our results showed that P effect size increased with the accumulated P addition amount and experimental duration (Fig. 3 and Supplementary Figs. 10 and 11). A more detailed description of data collection and the above justification are given in the revised Methods section (L282-287).

Another reason for selecting the measure with the highest P addition amount rather than the average of all P addition levels is that an error may be introduced during the calculation of the Standard Deviation (SD) of aboveground production from multiple SDs. We did examine the difference in the results between the two methods by weighting Ln(RR) uniformly. We found that the estimated effect sizes based on the mean of all P treatment levels in one experiment

(natural ecosystems: 35.0%, croplands: 15.0%) was a bit smaller than the estimates based only on the highest level of P addition (natural ecosystems: 36.3%, croplands: 16.3%). So, neither method would significantly affect our conclusions.

After removing the pseudo-replicates and the experiments with a basal fertilizer, our number of measures decreased from 1983 to 652. Despite this, our main conclusion about natural terrestrial ecosystems was unchanged and we have a new finding that P effect size in the croplands (13.9%) was much smaller than in the natural terrestrial ecosystems (34.9%). This new finding was discussed in the revised manuscript (L178-190).

Moreover, in the revision, we clearly defined what constitutes an experiment in this study, i.e., “We defined an experiment as a temporally and spatially distinct experiment with internally consistent controls ...” (L279-288).

Comment 3: #2 - The results of the Boost Regression Tree (BRT) are partly biased due to lack of data. The fact that the BRT method has a high tolerance to missing values in predictors (line 262) does not mean that this machine learning method is able to simulate the results without any information. In the Table S1, it is clear that most factors have a high percentage of non-missing values (except slope), which is OK. But, in the Table S2, it is also clear that the “parent material” factor has mainly missing values. Looking at the dataset, the proportion of missing values is 88% for the parent material. In such a case, this factor should be removed from the BRT analysis: one cannot determine if the small influence of parent material (Figure 4) is due to a real low importance of this factor or just a lack of data.

Response: Thanks for the critical comments. We agree that the relative importance of the variables with missing values would be underestimated by the BRT method. As suggested, parent material type was excluded from the BRT analyses in the revised manuscript (Fig. 4). Since some measurements such as soil available P and soil particle size are also missing, we have acknowledged this statistical problem in the discussion section (L243-246). Meanwhile, in the revised manuscript, we did not emphasize the relative importance of individual predictors (extended discussions were deleted), and instead, we stated “Climate, fertilization regimes, soil properties, and plant properties each contribute some portion (9.1%–40.0%) of the total explained variation in P effect size ...” (L206-209). We further proposed that “The multiple regulating factors of the P effect size rather than any single factor once more explained why P

limitation spread over the global land surface. For example,...” (L209-221).

Comment 4: #3 - Comparing croplands to natural ecosystems is interesting, but is incorrect. The croplands respond differently from natural ecosystems to P application (see Figure S8). Moreover, it is clear that the NKM class has a huge difference of response as compared with the other fertilization treatments (see Figure 3b). This enormous difference (one order of magnitude) could be explained by the supply of all nutrients (including micro-nutrients), alleviating all nutritional limitations (whereas, it is not the case for the other fertilization treatments). By the way, based on the dataset, I estimated that 92% of the NKM values are associated to croplands. The two following procedures should be done:

- The croplands should be analyzed separately from the other ecosystems. This is important when discussing the global importance of P-limitation (above all, in comparison with former studies; lines 110-113).
- The NKM treatment should be discarded because it is obviously different from the other treatments (Figures 3b, S8, and S9; see also lines 174-176 in SI).

Response: Thanks for the valuable comments and suggestions. As suggested, we have excluded the NKM treatment from the statistical analyses in the revised manuscript. We also excluded all the experiments with a basal fertilizer such as N, K, and NK to avoid any mixed effect. Indeed, 91% of the experiments with NKM as the basal fertilizers are from the croplands, and 78% of the experiments with a basal fertilizer are from the croplands. Therefore, removing the experiments with basal fertilizers significantly lowered the estimate of effect size in the croplands (decreased from 38.5% to 13.9%) but had a small influence on the estimate in the natural ecosystems (decreased from 37.0% to 34.9%).

Also, as suggested, we re-performed all statistical analyses separately for the natural ecosystems and the croplands (e.g., in Figs 1-4).

Moreover, we rechecked the previous estimates in natural ecosystems and made some corrections. Specifically, the four previous estimates in natural ecosystems were calculated using three different methods, i.e., Ln(RR) weighted uniformly as in the studies of Elser et al. (2007) and Augusto et al. (2017), Ln(RR) weighted by the inverse variance as in Yue et al. (2017), and RR weighted by the inverse variance as in Li et al. (2016). To make our results comparable with these studies, we estimated the global magnitude of P limitation using all the three methods and

found that our estimates in the natural ecosystems were consistently higher than the previous estimates (Table 1). We also corrected the number of observations in Yue et al. (2017) and Li et al. (2016). Yue et al. (2017) reported results of 143 observations from 60 experiments in main text and 60 observations from 60 experiments (removed pseudo-replicates) in Supplementary Information. Their estimate from the 60 observations is cited in our revised manuscript. We changed the number of experiments in the study of Li et al. (2017) from 135 to 50 (they reported results of 135 observations from 50 experiments).

Comment 5: Figure 1: there are not 1983 dots-experiments!

Response: As stated in Response 2, after discarding pseudo-replications and experiments with a basal fertilizer, now we have 652 measures from 652 experiments, which include 436 from natural ecosystems and 216 from croplands.

Comment 6: Lines 110-113: this comparison is biased (see above)

Response: Agreed and corrected. Please see the details in Response 4 and the new Table 1 in the revision.

Comment 7: Lines 116-117 and 119-122: incorrect. This applies only to Elser study. More recent studies (refs. 8-9) are, in practice, global.

Response: Thanks for point out this. In the revision, we corrected the comparisons. We showed the locations of the experiments included in the studies of Elser et al. (2007) and Augusto et al. (2017) and our study in the Supplementary Fig. 1 (data in Yue et al. (2017) and Li et al. (2016) were not available). Also, we proposed that the low effect size estimated in Elser et al. (2007) in comparison with our estimates was because most of their experiments are from Europe and North America (Supplementary Fig. 1b) where the magnitude of P limitation was generally smaller than the global average (Fig. 3; L165-172). The experiments in Augusto et al. (2017) are distributed very evenly over global land surface (Supplementary Fig. 1c). It is not exactly clear why the estimate by Augusto et al. (2017) is also smaller than our estimate. We assume that it may be because of a relatively small sample size in the previous meta-analyses (L172-177; Table 1; Supplementary Fig. 2), as indicated by our cumulative meta-analyses (Supplementary Fig. 4).

In the revision, we discarded the comparison between global N effect size and global P effect size (that was stated in previous version of our manuscript, L119-122).

Comment 8: Line 135 and Figure 4: the factor “parent material” should be discarded from the analysis because there are not enough data to calculate a reliable estimate (see above).

Response: Done.

Comment 9: Methods: I suggest the authors to quantify the quality of their meta-analysis using the scales of Koricheva (Koricheva & Gurevitch (2014). Uses and misuses of meta-analysis in plant ecology. *Journal of Ecology*, 102(4), 828-844.) or Philibert (Philibert et al. (2012). Assessment of the quality of meta-analysis in agronomy. *Agriculture, Ecosystems & Environment*, 148, 72-82.).

Response: Thanks for the great suggestion. As suggested, we checked the quality of our meta-analysis according to the checklist of Koricheva & Gurevitch (2014). After this revision, our meta-analysis fulfilled 15 of the 16 suggested quality criteria for a meta-analysis in plant ecology. The exception is the one that is not applicable to our study (i.e., check of the phylogenetic relatedness of species). In the revision, our quality check results are given in Supplementary Table 6. We also performed sensitivity tests using the cumulative meta-analysis and the leave-one-out meta-analysis (Supplementary Fig. 4), publication bias tests using funnel plot and relationship analysis (Supplementary Fig. 5), and tests of temporal change in P effect size using the meta-regression method (Supplementary Fig. 6). We have also added some discussions on the positive asymmetric distribution of P effect size in the croplands (L91-203). Moreover, we added the PRISMA flow gram as Supplementary Fig. 12 in the revision.

Comment 10: Lines 69-70 (SI): replace “latitude” by “altitude”.

Response: Corrected. This part of description has been moved to the Methods section (L341-343).

Comment 11: Line 167 (SI): replace “altitude” by “latitude”.

Response: The detailed discussion of latitudinal pattern has been discarded, as mentioned in Response 3.

Comment 12: Lines 183-184: this comparison is flawed (see above the first main issue).

Response: The discussion has been discarded due to the flaw.

Comment 13: Figure S1: difficult to distinguish the croplands from the tundra. Please change the colors.

Response: The figure (Supplementary Fig. 1a) has been redrawn with larger and distinct symbols and more distinguishable colors.

Comment 14: Figure S3d: this comparison is potentially biased because Eucalyptus species are generally found in tropical and subtropical areas while Pinus species are more frequent in boreal and temperate region than in tropical-subtropical regions.

Response: The comparison has been discarded due to the potential bias.

Comment 15: Figure S9a: difficult to distinguish the class “slightly” from the class “Intermediately”. Please change the colors.

In addition, replace “lightly” by “slightly”.

Response: The figure has discarded, since we removed the detailed discussion on latitudinal pattern.

Response to Reviewer 2

Comment 1: This is a well written and interesting manuscript.

Response: Thanks the reviewer for the positive comments on our study.

Comment 2: However, I do not think that the assertions made about the novel, insightful nature of the work are fully justified. The role of phosphorus limitations has been explored in many previous publications particularly in agricultural management and agronomy.

Response: We totally agree that the role of P supply has been explored in a number of studies, particularly in croplands (at least since the Rothamsted research in 1840s). In other words, we've known that P is important for plant growth for nearly two centuries. However, the question remains unknown is how important P is for plant production (i.e., the magnitude of P limitation) and where P limits plant production at the global scale. These are the questions we are addressing in this study. This is critical for biological conservation and the prediction of terrestrial C sequestration potential and has implications for croplands managements.

Moreover, our focus is on natural terrestrial ecosystems (we made this point clearer in the revision by adding “natural” before “terrestrial ecosystems” in the revised title). Despite a long history of P study, Elser et al. (2007) may be the first one that examined the global magnitude of P limitation in natural terrestrial ecosystems, which included 107 terrestrial P addition experiments. Recently, there are three more global meta-analyses that focused on N-P interactions (i.e., Augusto et al. (2017), Yue et al. (2017), Li et al. (2016)). The three recent meta-analyses had a sample size (50-117) comparable to that in Elser et al. (2007), although lots of P addition experiments had been published since 2007. The overwhelming motivation of this study was that with more experiments published would the conclusion drawn by previous meta-analyses with 50-117 experiments be changed? An example of ‘conclusion changed with sample size’ can be seen in a recent paper on soil organic C (van Gestel et al., Nature, 2018). With a much larger dataset in natural terrestrial ecosystems (3.8-8.8 times of those in previous meta-analyses), we found a more widespread (significant in all climate regions and major types of ecosystems) and much (7.0%-15.9%) stronger P limitation in natural terrestrial ecosystems than previously suggested (Fig. 1 and Table 1). Moreover, we found that the P effect size increased with experimental duration and P addition amount. After weighted with experimental duration or

P addition amount, the P effect size was even larger (4.2%-12.1% larger). These results highlight an underestimated role of altered P supply on terrestrial plant production.

As commented by Reviewer 1, “although several publications already gave such a global perspective (refs. 1, 8, 23), the present manuscript is welcomed because the reference 1 needs an update, and because the references 8 and 23 focused on nitrogen (N)-P interaction (implying a smaller dataset as all “P-alone” experiments could not be taken into account). In addition, as opposed to former publications, this study also includes the croplands (and not only the “natural” ecosystems, such as forests, wetlands, and so on).” and “The authors did a good job in collecting many original data for their meta-analysis. In meta-analyses, it is almost impossible to claim for data exhaustiveness. But, in this case, the amount of collected data is excellent.” Not only more data are provided, novel notion is also provided, that is pervasive phosphorus limitation in terrestrial ecosystems.

Even though our sample size in natural terrestrial ecosystems was 3.8 to 8.8 times of those in the previous syntheses, there is still a large uncertainty in our estimated P effect size (confidence interval of 30.0%–40.1%). Many ecosystems remain underrepresented in our global database, e.g., ecosystems in North Asia and the tropics and mature mixed forests. Therefore, global magnitude and distribution of P limitation are still far from well understood. Our study represents a significant step forward in this perspective, which will “interest a large audience in the scientific community” as Reviewer 1 had commented. Moreover, surprisingly, we didn’t find any meta-analysis on P addition experiments in croplands at the global scale, although there are several ones at the country level (e.g., in Finland; listed in our Dataset).

Changes made: In order to make our study more distinct from previous studies, we have made the following changes in the revision: (1) We added “natural” before “terrestrial ecosystems” in the title to highlight that our focus is on natural terrestrial ecosystems. (2) We justified the necessity of a larger dataset: “This most up-to-date dataset better capture Earth’s diverse terrestrial habitats, enabling a much clearer understanding of the role of P supply on terrestrial plant production.” (L53-54). (3) Global distribution of terrestrial P addition experiments in previous syntheses were given in Supplementary Fig. 1, as a comparison to the site locations in our study. (4) A comparison of the accumulated number of experiments with time in this and previous meta-analyses were given as Supplementary Fig. 2. As shown in Supplementary Fig. 2 and described in L47-50, 41.7% of our collected experiments were published since 2007. (5) We

provided details of the comparison between our estimates and the previous estimates in Table 1. (6) We discussed the uncertainties in our estimate and how future work can improve the estimates at the end of our discussion (L229-248).

Reference

1. van Gestel, N. et al. Predicting soil carbon loss with warming. 554, E4 (2018).

Comment 3: Furthermore, I was not convinced that combining agricultural and 'natural' ecosystems was relevant or technically sounds. The removal of biomass (and phosphorus) with crop harvest leads to entirely different biogeochemical cycling pathways. Therefore, I do not recommend this manuscript for publication in this journal.

Response: Thanks for the critical comment. We agree that combining agricultural and natural ecosystems might make it difficult to interpret the results, since they have very different biogeochemical cycling pathways. To address the reviewer's concerns, in the revision we re-performed all statistical analyses separately for croplands and natural terrestrial ecosystems. Our previous conclusion still holds in natural terrestrial ecosystems while we have a new result in croplands: P limitation in croplands is much smaller than in natural terrestrial ecosystems. The estimate in croplands is now lower due mainly to the removal of experiments with a basal fertilizer. The new result was discussed in L178-190.

Nevertheless, we think P addition experiments in croplands and natural ecosystems are relevant and a comparison between them can provide new insights though traditionally they are separated. Both the concept of nutrient limitation and the traditional approaches to identify nutrient limitation (ideally using nutrient addition experiments) have been created for croplands and were later applied to natural ecosystems. Understanding nutrient limitation in natural ecosystems has implications for nutrient managements in croplands. Croplands are initially natural terrestrial ecosystems and there are still many natural terrestrial ecosystems being converted to croplands (e.g. in Amazon). Including both natural terrestrial ecosystems and croplands better capture the Earth's diverse terrestrial habitats.

Response to Reviewer 3

Comment 1: The authors assemble data on plant growth responses to P fertilization and perform a meta-analysis to assess the extent of P limitation worldwide. Understanding this is critical given the anticipated role of nutrients in mediating plant responses to environmental change, especially increasing atmospheric carbon dioxide concentrations globally. In that sense, this research addresses an important topic. However, I think there are a number of shortcomings with the current version of the paper that should be addressed before it is published. I would argue that at least in its current form, the paper should not be considered for publication in Nature Communications. The authors can find my comments below.

Response: Many thanks for the generally positive comments and suggestions that helped to improve our manuscript. We've addressed all the concerns the reviewer raised in the revision, as explained below.

Comment 2: The authors should be cautious with terminology. For example, the paper argues that this analysis includes 1983 “fertilization experiments” in the abstract and elsewhere. I can't tell for sure from the data set, but the number of independent experiments seems to be far fewer than that. To me, it would seem more appropriate to actually analyze different experiments rather than treating individual results as separate, independent data.

Response: Thanks for the suggestion. In the revision, we clearly defined an experiment as “a temporally and spatially distinct experiment with internally consistent controls” (L279-280). Previously, multiple nutrient addition levels of an experiment were included. In the revision, only the treatment with the highest amount of P addition in each experiment was retained. The justification for this selection is “Choosing the latest measure and the highest P addition amount increased the likelihood that P additions fulfill plant demand and overcome the sorption of P fertilizer by soils and soil microbial competition for P fertilizer” (L285-287). After discarding non-independent data, the sample size decreased from 1983 to 1023. The sample size was further decreased to 652 after excluding experiments with a basal fertilizer.

Comment 3: There is a great deal of confusion (and in some cases contradictory information) from the literature about the nature of nutrient limitation in different sites because some studies

only look at the responses to single nutrient addition. For example, like this study did for P, LeBauer and Treseder (2008) only looked at plant responses to N, even when response data may have been available for both nutrients and clearly show that both are important. It has been clearly established that in many cases, co-limitation (by N and P) is the rule (Elser et al. 2007; Harpole et al. 2011). I think the authors should be both cognizant of that, and explicitly address that issue in the manuscript. Without combing through the data from all of these papers, I would suspect there are sites where plants respond to both N and P, but suggesting that those sites are (only) limited by a single nutrient adds confusion to our understanding about what the nature and extent of limitation really is. I think this needs to be addressed in the paper.

References:

- LeBauer, D. S., & Treseder, K. K. (2008). Nitrogen limitation of net primary productivity in terrestrial ecosystems is globally distributed. *Ecology*, 89(2), 371–379.
- Elser, J. J., Bracken, M. E. S., Cleland, E. E., Gruner, D. S., Harpole, W. S., Hillebrand, H., ... Smith, J. E. (2007). Global analysis of nitrogen and phosphorus limitation of primary producers in freshwater, marine and terrestrial ecosystems. *Ecology Letters*, 10(12), 1135–1142.
- Harpole, W. S., Ngai, J. T., Cleland, E. E., Seabloom, E. W., Borer, E. T., Bracken, M. E. S., ... Smith, J. E. (2011). Nutrient co-limitation of primary producer communities. *Ecology Letters*, 14(9), 852–862.

Response: Thanks for the critical comments. A primary motivation of this study (i.e., to focus on P addition experiments) is that we have known much about N limitation from N addition experiments (e.g., LeBauer and Treseder, 2008) as well as NP co-limitation from full factorial NP addition experiments (e.g. Harpole et al. 2011). But we rarely systematically looked into P addition experiments, and P addition alone experiments have usually been ignored in previous meta-analyses. As N limitation and dynamics have been much better known than that of P, this study was aimed to advance our understanding of P limitation through a data synthesis. We looked into whether plant growth at a site is responsive to P addition or not. Please note that a site with P limitation does not necessarily mean it is not limited by other nutrients. We have carefully checked the whole manuscript to not state that the sites are only limited by P supply. Actually, our opinion on nutrient limitation is that, P limitation and N limitation usually occur together on almost all lands, and it's a matter of the magnitude of limitation.

To avoid any confusion as concerned by this reviewer, we have excluded the experiments with any basal fertilizers such as N, K, and NK in the revision. We looked into the difference only between the control (no fertilizer added) and the highest P addition treatment (only P fertilizer was added). All results and discussion in the revision are based on this sub-dataset.

Comment 4: Combining the responses of agricultural and natural systems may be providing a misleading picture of nutrient limitation globally. Ag systems are often single species systems (and well-watered), whereas natural systems have much more diversity and where climate conditions may attenuate nutrient addition responses. Clearly the responses to nutrient additions would be expected to be greater in a monotypic, watered system, and those responses may be biasing the “average” responses reported in the paper. I would suggest that the authors include an analysis of both natural and ag systems separately to examine this potential bias. Short of that, I worry that this paper may only add more confusion to the issue.

Response: Thanks for the kind suggestion. As suggested, we re-performed all statistical analyses separately for the croplands and the natural ecosystems in the revision. The revised analyses showed that our previous statement (a more widely spread and much stronger P limitation) holds true for the natural terrestrial ecosystems. However, the croplands exhibited a much smaller magnitude of P limitation than the natural terrestrial ecosystems, practically as a result of the exclusion of the experiments with a basal fertilizer. We attributed the difference between the two types of ecosystems mainly to the possible historical fertilizations in the croplands (L182-190), as indicated by the relatively high soil extractable P concentration in the croplands compared to the natural terrestrial ecosystems (Supplementary Fig. 8).

We also expected single species systems to be more responsive to P additions than species-rich systems. According to our results, this is true for forest ecosystems, as shown in the new Supplementary Fig. 7. However, the difference between the croplands and the natural terrestrial ecosystems may be mainly caused by their difference in soil P availability, rather than species composition.

Comment 5: The authors should more clearly articulate how “plant production” was determined, especially in the natural systems. The database includes “mean shoot production,” but how was this determined for forests? Having worked in forests my whole career, I know that it is at least

as common for studies to assess just litterfall. How was this addressed here? As is, there is simply far too little detail in the methods for the reader to assess what the results mean and how they were obtained.

Response: Thanks for the comment. We agree that how “plant production” was determined is critical for assessing the meaning of the results. While researchers are usually interested in the production of whole plants, the aboveground production rather than the production of whole plants is more often assessed. Therefore, most measures in our database are aboveground production (details in Supplementary Table 5). To make this point clear, we carefully rechecked our dataset with a focus on forest ecosystems and added more descriptions of the data in the revision. To be consistent, we changed both “plant production” and “mean shoot production” to “aboveground production” for our dataset.

Measures are relatively simple for ecosystems other than forests. In the revision, we gave detailed information on the measures in all types of ecosystems in the Supplementary Table 5 and our enclosed dataset. In forest ecosystems, measures included are aboveground biomass production ($N = 33$, typically calculated as the difference in biomass between two time points), and the increase rate of diameter (34), height (16), basal area (25), and volume (25). We did not find any significant difference in P effect size among these measures (Supplementary Table 5). Only 1 measure of litterfall production in our database. This is because litterfall production was not measured as commonly as diameter in the P addition experiments and also because we preferred measures of the production of the whole aboveground part over litterfall production when both are available. Litterfall production usually only measures the production of leaf and trig, not including the production of woody part. The diverse measures may cause some uncertainty in our estimate as acknowledged in our revised discussion (L242-243), but we preferred to include all these measures to better represent the Earth’s diverse natural habitats.

Comment 6: Line 141. Basal applications of N, K, and micronutrients? This is a huge issue I think that needs to be discussed more. how many of the studies fall into this category? I would consider removing these from the analysis, as previous additions of other potentially limiting nutrients could simply be “baiting” a P response via stoichiometric imbalances.

Response: Thanks for the comments. As suggested, we removed all experiments with any basal fertilizer from the revision. Most experiments with a basal fertilizer are from croplands.

Therefore, discarding experiments with a basal fertilizer reduced the P effect size in croplands but had a minor influence on the estimate in natural terrestrial ecosystems.

Comment 7: Line 171: Yet, the message here is that the world is P limited, and that does not bode well for careful P management without some careful caveats. Given that, I think some of the analyses I suggest above are critical to bolster the results, and especially the conclusions.

Response: Thanks for the comment. We agree our previous results may mislead P management in croplands. In the revision, we have excluded all experiments with a basal fertilizer. Now the estimated P limitation magnitude in croplands is much smaller (13.9%), probably as a result of historical fertilizations in croplands.

Reviewers' comments:

Reviewer #1 (Remarks to the Author):

I want to commend the authors for their very good work. Indeed, they have improved a lot the quality of their study, which is now excellent. I am convinced that this meta-analysis will be highly useful for the scientific community.

Only a few minor changes are still needed before publication:

- Line 21: replace "301 out of the 652" by "almost half (46.2%) of the"
- Line 27: insert a sentence before "Our results [...]" that explains that P limitation is found in non-tropical regions (in link with the first sentence of the Abstract). For instance: "While our study confirmed that P limitation is widespread in tropical regions, it demonstrated that other regions are often also P-limited.", or any other rephrase of this idea.
- Line 51: insert "dedicated to N-P interactions" after "syntheses"
- Line 61: explain that "ln" is the natural logarithm
- Line 69: "occurring in all studied continents (Figs. 1, 2, and 3)."
- Line 156: remove "considerably"
- Lines 173-175: it might be because they used a slightly different way of removing the pseudo-replications (in SI Methods): "[...], when plant response was monitored at several dates after the treatment date, we used only one date, choice being based on the type of ecosystem and on recommendations made by Sullivan et al. (2014): the latest date of biomass measurement for standing forest biomass [...] or the earliest date of measurement for other ecosystems [...]".
- Lines 183-188: I agree. Agricultural soils in Europe have been enriched in phosphorus over decades. See Ringeval et al. (2014, *Global Biogeochemical Cycles*, 28(7), 743-756) who demonstrated that up to 90% of soil P could originate from former fertilization applications.
- Lines 295-296: Unclear. Please rephrase.
- Lines 341-342: there are not five belts but four (in absolute latitude values) or seven (in positive or negative values).
- Figure S8: is this relationship still significant when croplands are removed? It should be tested.
- Figure S12: good. But I don't understand what is the "qualitative synthesis". Please explain or remove.

Reviewer #2 (Remarks to the Author):

I appreciate the additions and improvements made to the manuscript. I found myself with two basic questions when reading this manuscript again.

1. what are you basing your soil input variables (available P etc.) on? Is it the no P treatment?
2. Given that you state that Cropland P response has been mitigated by prior P fertilization - why does it make sense to aggregate across sites and regions with wildly different fertilization histories.
3. Is this information going lead to greater understanding of global P budgets and inform earth system models?

As a cropland scientist, I don't find myself gaining much information from this exercise.

Reviewer #3 (Remarks to the Author):

Overall, the authors have done a reasonable job responding to reviewer comments. I do understand the choice to focus on P (alone), but I stand by my comment that by doing so, we have not really advanced our understanding of the true nature of nutrient limitation. If this is published, we will be left with two papers, one that argues that the world is strongly and mostly N limited, one that argues that it is strongly and mostly P limited. For much of the world, co-limitation is probably most important (Harpole et al. 2011), but this does not address that, and thus I fear this will only create more confusion. At the very least, i would hope the authors would address that head on in this manuscript.

Below I offer a number of specific comments below that I hope the editor/authors will consider prior to accepting this article for publication.

Line 33. While I agree that the most focused on patterns of limitation have argued that N limitation rules in the temperate zone and P limitation in the tropics, more recent research has clearly shown that latitude alone is not the driver. What really matters is weathering, and much of our most compelling work has been done along chronosequences, where, independent of latitude, time since disturbance is most important. In climatically stable areas outside the tropics, we also see strong evidence of P limitation in highly weathered sites (E.g., Wardle et al. 2004; Peltzer et al. 2010; Laliberte et al. 2012).

Line 37: ABOVEGROUND plant production. This is an important clarification, as some of the increases in ANPP with fertilization may reflect changes in carbon allocation (from below to aboveground) rather than increases in PLANT NPP per se.

Line 43. It would be worth noting here or in the discussion that given the span of data used (since 1950), the nature of and strength of nutrient limitation have likely changed over much of the land covered in this analysis. See Penuelas et al. increases in N availability, for example, have likely driven stoichiometric imbalances that would be hypothesized to enhance P limitation.

Line 54: please be specific throughout: aboveground plant production.

Line 55: Please define "threshold value of P limitation."

Line 60 - 61: I am not following this. Typically, RR is often calculated as \ln of the treatment/control (e.g., Elser et al. 2007). For studies that reported RR differently, how were the responses standardized? I also think more detail on the weighting approach is needed, as this brief description will be lost on most readers (i.e., weighted by the inverse variance).

Line 68: data should be plural (no data were available).

Line 71: Does the 46.2% value apply to all ecosystems, or to ag or natural? If the former, I suggest you also include the proportions of ag and natural.

Fig. 2 legend: Instead of focusing on latitudes, I would just state that P limitation occurred broadly across most/all ecosystems. The extent of P limitation likely reflects differences in the degree of soil weathering, which occurs across latitude, but also can vary independent of latitude.

Line 131: can you please report stocks as mg/kg or something similar? g/m²/year is not a very meaningful unit for a P stock (ie. it varies based on soil depth, bulk density, etc). Also, this number is reported as a flux, not a stock. Perhaps I am just not understanding, but either way, please clarify.

Line 132: chronic losses of P over very, very long timescales.

Line 134: This ignores perhaps the most important mechanism: P becomes “fixed” in chemically and or physically “occluded” forms. Many tropical forests have plenty of total P, it’s just mostly in forms that many plants can’t easily access.

Line 136: Soil barriers? What does this mean? We know enough about P biogeochemistry in soils to avoid being vague like this.

Line 138: other resources broadly, yes, but particularly CO₂ and N, from a stoichiometry perspective.

Line 141: No. N fixers are not necessarily abundant in “cold areas” as this statement applies. And the presence of putative fixers does not guarantee fixation is occurring. There are some pretty fundamentally incorrect/misleading/vague statements about N and P biogeochemistry throughout this section that should be addressed.

Line 144: In natural ecosystems (remove “the”)

Line 151: “to compare with three other”

Table 1 Legend: “The magnitude of P limitation in natural terrestrial ecosystems is larger than previously estimated” Also “experiment” should be “experiments” in columns 3 and 5.

Line 165: “compared to our study”

Line 175: Also important to note that the Elser data included studies that were natural systems, but also phytometer/plantation data as well.

Line 195: What does this mean: Therefore, there is no apparent tendency to 195 publish a statistically significant P effect. This is not clear. Please explain.

There are actually quite a few minor grammatical errors throughout. I would suggest a very careful read/edit by a native English speaker/editor prior to publication.

Line 258: this is a disappointing way to end the paper. I don’t disagree with the statement, but it comes out of nowhere, and the context for it is not clear. I would either make it meaningful (explain) or omit it.

Harpole, W. S., & al., et. (2011). Nutrient co-limitation of primary producer communities. *Ecology Letters*, 14, 852–862.

Peltzer, D. A. et al. (2010). Understanding ecosystem retrogression. *Ecology*, 80(4), 509–529.

Wardle, D. a, Walker, L. R., & Bardgett, R. D. (2004). Ecosystem properties and forest decline in contrasting long-term chronosequences. *Science (New York, N.Y.)*, 305(5683), 509–513.

Laliberte, E., & al., et. (2012). Experimental assessment of nutrient limitation along a 2-million-year dune chronosequence in the south-western Australia biodiversity hotspot. *Journal Of Ecology*, 100, 631–642.

Response to Reviewer 1

Comment 1: I want to commend the authors for their very good work. Indeed, they have improved a lot the quality of their study, which is now excellent. I am convinced that this meta-analysis will be highly useful for the scientific community.

Response: Many thanks for the positive comments.

Only a few minor changes are still needed before publication:

Comment 2: - Line 21: replace “301 out of the 652” by “almost half (46.2%) of the”

Response: Done (L21-22).

Comment 3: - Line 27: insert a sentence before “Our results [...]” that explains that P limitation is found in non-tropical regions (in link with the first sentence of the Abstract). For instance: “While our study confirmed that P limitation is widespread in tropical regions, it demonstrated that other regions are often also P-limited.”, or any other rephrase of this idea.

Response: Done (L28-30).

Comment 4: - Line 51: insert “dedicated to N-P interactions” after “syntheses”

Response: Done (L56-57).

Comment 5: - Line 61: explain that “ln” is the natural logarithm

Response: Done (L69).

Comment 6: - Line 69: “occurring in all studied continents (Figs. 1, 2, and 3).”

Response: Done (L79-80).

Comment 7: - Line 156: remove “considerably”

Response: Done (L167).

Comment 7: - Lines 173-175: it might be because they used a slightly different way of removing the pseudo-replications (in SI Methods): “[...], when plant response was monitored at several dates after the treatment date, we used only one date, choice being based on the type of ecosystem and on recommendations made by Sullivan et al. (2014): the latest date of biomass measurement for standing forest biomass [...] or the earliest date of measurement for other ecosystems [...]”.

Response: This explanation has been incorporated into the revised manuscript (L184-187).

Comment 8: - Lines 183-188: I agree. Agricultural soils in Europe have been enriched in phosphorus over decades. See Ringeval et al. (2014, Global Biogeochemical Cycles, 28(7), 743-

756) who demonstrated that up to 90% of soil P could originate from former fertilization applications.

Response: Thanks for the suggestion. Ringeval et al. (2014) has been added as an example of P enrichment in croplands in the revised manuscript (L198-200).

Comment 9: - Lines 295-296: Unclear. Please rephrase.

Response: Done (L337-338).

Comment 10: - Lines 341-342: there are not five belts but four (in absolute latitude values) or seven (in positive or negative values).

Response: Corrected (L385).

Comment 11: Figure S8: is this relationship still significant when croplands are removed? It should be tested.

Response: The suggested analysis has been performed during the revision. If croplands are removed, the relationship becomes non-significant ($P = 0.11$, $R^2 = 0.07$, $N = 39$). This information has been added to the caption of Figure S8.

Comment 12: - Figure S12: good. But I don't understand what is the "qualitative synthesis". Please explain or remove.

Response: Agreed. This phase is not clear and has been removed.

Response to Reviewer 2

Comment 1: I appreciate the additions and improvements made to the manuscript. I found myself with two basic questions when reading this manuscript again.

1. what are you basing your soil input variables (available P etc.) on? Is it the no P treatment?

Response: We thank the reviewer very much for the positive comments and further questions. As modified in L361-363: “Site characteristics included ... soil physiochemical properties before the experiments began or from the control treatments (concentrations of available P, organic C, and total N; pH in water; and particle size).”

Comment 2: 2. Given that you state that Cropland P response has been mitigated by prior P fertilization - why does it make sense to aggregate across sites and regions with wildly different fertilization histories.

Response: Thanks. Please note that we did not know whether crop P response has been mitigated by possible prior P fertilization before we perform this study. This is a hypothesis we proposed, based on two of our results: the generally lower magnitude of P limitation and the generally higher soil available P concentration in croplands compared to natural terrestrial ecosystems (L190-195). One of our criteria to include an experiment in our database is “no fertilization was recorded in the control treatment either before the start of the experiment or during the experiment” (L317-318). In other words, the studied sites might have a fertilization history before the experiments, but the researchers did not mention it in their papers.

The reason we have aggregated experiments across sites and regions is to generate a global perspective of the nature of P limitation in croplands. In the revision, we realized that cropland P limitation may change with time and fertilization history. Therefore, we added a note in the discussion section: “given the long span of time of the datasets (Supplementary Fig. 2), the nature of nutrient limitation has likely changed over much of the land covered in this analysis, due to the changes in atmospheric N deposition and to changes in fertilization practices in the croplands.” (L284-287).

Comment 3: 3. Is this information going lead to greater understanding of global P budgets and inform earth system models?

Response: Regarding P budgets and modelling, our results imply that “Our results therefore suggest that P limitation in croplands has been largely alleviated by historical fertilizations^{30, 31} and that a reduced amount of P fertilizer is needed to increase crop production in the future. It follows that to accurately predict the future fertilizer effect on crop production, models require fertilization history.” These implications have been added in the revised discussion section (L200-203).

Comment 4: As a cropland scientist, I don't find myself gaining much information from this exercise.

Response: The key contribution on croplands by this meta-analysis is to gain a global perspective of P limitation on crop production, i.e., a smaller magnitude of P limitation due to the generally higher soil P availability compared to natural terrestrial ecosystems. This analysis includes both both natural terrestrial ecosystems and croplands, which together covers over 90% of the global land surface, and can help understand P limitation at the global scale.

Response to Reviewer 3

Comment 1: Overall, the authors have done a reasonable job responding to reviewer comments. I do understand the choice to focus on P (alone), but I stand by my comment that by doing so, we have not really advanced our understanding of the true nature of nutrient limitation. If this is published, we will be left with two papers, one that argues that the world is strongly and mostly N limited, one that argues that it is strongly and mostly P limited. For much of the world, co-limitation is probably most important (Harpole et al. 2011), but this does not address that, and thus I fear this will only create more confusion. At the very least, I would hope the authors would address that head on in this manuscript.

Harpole, W. S., & al., et. (2011). Nutrient co-limitation of primary producer communities. *Ecology Letters*, 14, 852–862.

Response: We thank the reviewer for the positive comments and further suggestions on our manuscript. We totally understand the reviewer's concern about the whole picture of nutrient limitation. To address this concern, we added a paragraph to the revised manuscript (L245-265) to discuss this issue.

In general, although our results and previous findings seem conflicting (i.e., globally distributed P limitation by this study, N limitation by LeBauer and Treseder (2008), and N and P co-limitation by Elser et al. (2007) and Harpole et al. (2011)), they can be reconciled by the multiple limitation hypothesis. A prevalent co-limitation of terrestrial primary production by N and P suggests a generally balanced N and P limitation in global terrestrial ecosystems (Elser et al., 2007; Harpole et al., 2011), while a global distributed N limitation indicates widespread N limitation in terrestrial ecosystems (LeBauer and Treseder, 2008). According to these conclusions, a worldwide spread of P limitation in terrestrial ecosystems is expected, as observed in the present study; the absence of a worldwide occurrence of P limitation would either imply an imbalance of N and P limitation, which would counter the finding of prevalent N and P co-limitation (if widespread N limitation is true), or imply a less widespread N limitation, which would counter the finding of globally distributed N limitation (if prevalent N and P co-limitation is true). In brief, worldwide spread of P limitation actually matches with and can be expected from worldwide spread N limitation and prevalent N and P co-limitation, instead of contrary with them. The worldwide spread P limitation as identified in this study and the worldwide spread N limitation reported in LeBauer and Treseder (2008) together imply that P limitation and N limitation are largely independent of each other. This possibility is supported by another synthesis study, which reported that the effects of N supply and P supply on aboveground plant production were additive in most terrestrial ecosystems (Yue et al. 2017).

Below I offer a number of specific comments below that I hope the editor/authors will consider prior to accepting this article for publication.

Comment 2: Line 33. While I agree that the most focused on patterns of limitation have argued that N limitation rules in the temperate zone and P limitation in the tropics, more recent research

has clearly shown that latitude alone is not the driver. What really matters is weathering, and much of our most compelling work has been done along chronosequences, where, independent of latitude, time since disturbance is most important. In climatically stable areas outside the tropics, we also see strong evidence of P limitation in highly weathered sites (E.g., Wardle et al. 2004; Peltzer et al. 2010; Laliberte et al. 2012).

Peltzer, D. A. et al. (2010). Understanding ecosystem retrogression. *Ecology*, 80(4), 509–529.

Wardle, D. a, Walker, L. R., & Bardgett, R. D. (2004). Ecosystem properties and forest decline in contrasting long-term chronosequences. *Science (New York, N.Y.)*, 305(5683), 509–513.

Laliberte, E., & al., et. (2012). Experimental assessment of nutrient limitation along a 2-million-year dune chronosequence in the south-western Australia biodiversity hotspot. *Journal Of Ecology*, 100, 631–642.

Response: Thanks for pointing out the critical role of soil weathering in determining P limitation. We agree with the comments. To include the information provided by the reviewer, we added “For example, significant P limitation on aboveground plant production has also been found in some temperate areas with strongly weathered soils^{10, 11, 12}.” in L38-40.

Comment 3: Line 37: ABOVEGROUND plant production. This is an important clarification, as some of the increases in ANPP with fertilization may reflect changes in carbon allocation (from below to aboveground) rather than increases in PLANT NPP per se.

Response: “aboveground plant production” has been consistently used for describing our datasets in the revision.

Comment 4: Line 43. It would be worth noting here or in the discussion that given the span of data used (since 1950), the nature of and strength of nutrient limitation have likely changed over much of the land covered in this analysis. See Penuelas et al. increases in N availability, for example, have likely driven stoichiometric imbalances that would be hypothesized to enhance P limitation.

Response: Thanks for the suggestion. This point has been noted during the discussion of the uncertainties of our estimates in the revision in L284-287: “given the long span of time of the datasets (Supplementary Fig. 2), the nature of nutrient limitation has likely changed over much of the land covered in this analysis, due to the changes in atmospheric N deposition³⁸ and to changes in fertilization practices in the croplands³⁰”.

Comment 5: Line 54: please be specific throughout: aboveground plant production.

Response: Done.

Comment 6: Line 55: Please define “threshold value of P limitation.”

Response: We have defined it as “a critical P effect size best corresponded to a critical z-score at $P = 0.05$ ” in the revision (L64).

Comment 7: Line 60 - 61: I am not following this. Typically, RR is often calculated as \ln of the treatment/control (e.g., Elser et al. 2007). For studies that reported RR differently, how were the responses standardized? I also think more detail on the weighting approach is needed, as this brief description will be lost on most readers (i.e., weighted by the inverse variance).

Response: Yes, we agree that RR is often calculated as \ln of the treatment/control (e.g., Elser et al., 2007). However, there are some studies calculated RR as the treatment/control ratio (e.g., Li et al., 2016). To make our results comparable with these previous studies and to make the two methods distinguishable, we used RR to represent “response ratio” and used $\ln(\text{RR})$ to represent “ \ln transformed response ratio” in our study. Clearly, our abbreviations are better than the typical one (i.e., use RR to represent “ \ln transformed Response Ratio” rather than “Response Ratio”). In Li et al. (2016), RR is calculated as the treatment/control ratio and weighted by the inverse variance method (Table 1). In Elser et al. (2007), $\ln(\text{RR})$ is weighted uniformly, i.e., the same weight for all experiments (Table 1). Actually, $\ln(\text{RR})$ weighted by the inverse variance method is the most recommended approach to do meta-analysis in ecological studies (Gurevitch et al., 2018).

We have very detailed descriptions of the weighing approaches used in our study in the Methods (L431-440), including the description of inverse variance method in L434-440. To note this, we added “(details in Methods)” after the description of “weighted by the inverse variance” (L70-71). Please note that “inverse variance” is a commonly used term in meta-analysis, as seen in Gurevitch et al., (2018).

Reference

Gurevitch, J., Koricheva, J., Nakagawa, S. & Stewart, G. Meta-analysis and the science of research synthesis. *Nature* 555, 175-182 (2018).

Comment 8: Line 68: data should be plural (no data were available).

Response: Corrected (L78).

Comment 9: Line 71: Does the 46.2% value apply to all ecosystems, or to ag or natural? If the former, I suggest you also include the proportions of ag and natural.

Response: It’s for the former. As suggested, the proportions in croplands (48.6%) and natural terrestrial ecosystems (45.0%) have been added (L81-82).

Comment 10: Fig. 2 legend: Instead of focusing on latitudes, I would just state that P limitation occurred broadly across most/all ecosystems. The extent of P limitation likely reflects differences in the degree of soil weathering, which occurs across latitude, but also can vary independent of latitude.

Response: Done as suggested (L98).

Comment 11: Line 131: can you please report stocks as mg/kg or something similar? g/m²/year is not a very meaningful unit for a P stock (ie. it varies based on soil depth, bulk density, etc).

Also, this number is reported as a flux, not a stock. Perhaps I am just not understanding, but either way, please clarify.

Response: Sorry for the confusion. Here we mean a flux rather than a stock. We added “rate” after “soil P supply” in the revision (L142).

Comment 12: Line 132: chronic losses of P over very, very long timescales.

Response: As suggested, “chronic” is added before “loss of P” (L144).

Comment 13: Line 134: This ignores perhaps the most important mechanism: P becomes “fixed” in chemically and or physically “occluded” forms. Many tropical forests have plenty of total P, it’s just mostly in forms that many plants can’t easily access.

Response: Thanks for pointing out this mechanism. This mechanism has been added in the revision (L144).

Comment 14: Line 136: Soil barriers? What does this mean? We know enough about P biogeochemistry in soils to avoid being vague like this.

Response: In the revision, “soil barriers” has been changed to “the formation of soil layers (e.g., iron pans) that physically prevent/inhibit access by roots to potentially available P” (L147-149), as explained in Vitousek et al. (2010).

Reference: Vitousek, P. M., Porder, S., Houlton, B. Z. & Chadwick, O. A. Terrestrial phosphorus limitation: mechanisms, implications, and nitrogen-phosphorus interactions. *Ecol. Appl.* 20, 5-15 (2010).

Comment 15: Line 138: other resources broadly, yes, but particularly CO₂ and N, from a stoichiometry perspective.

Response: We agree and “especially N and atmospheric CO₂” has been added (L151).

Comment 16: Line 141: No. N fixers are not necessarily abundant in “cold areas” as this statement applies. And the presence of putative fixers does not guarantee fixation is occurring. There are some pretty fundamentally incorrect/misleading/vague statements about N and P biogeochemistry throughout this section that should be addressed.

Response: This example has been removed from the revised manuscript.

Comment 17: Line 144: In natural ecosystems (remove “the”)

Response: Done (L156).

Comment 18: Line 151: “to compare with three other”

Response: Corrected (L164).

Comment 19: Table 1 Legend: “The magnitude of P limitation in natural terrestrial ecosystems

is larger than previously estimated” Also “experiment” should be “experiments” in columns 3 and 5.

Response: Done (Table 1).

Comment 20: Line 165: “compared to our study”

Response: “than in our study” has been added (L176).

Comment 21: Line 175: Also important to note that the Elser data included studies that were natural systems, but also phytometer/plantation data as well.

Response: Thanks for noting this. Our synthesis, Elser et al. (2007), and also some other previous syntheses (e.g., August et al., 2017) all included plantations. Similarly, all the studies (including our study) select field/*in situ* experiments for research. We have made this point clear in the revised Methods (L339-340): “; natural forests, plantations, shrublands, and savannas were all classified as forest”.

Comment 22: Line 195: What does this mean: Therefore, there is no apparent tendency to 195 publish a statistically significant P effect. This is not clear. Please explain.

Response: The sentence has been rephrased to “Therefore, there is no apparent tendency of journals to favor publication of statistical significant P effects.” (L211-212).

Comment 23: There are actually quite a few minor grammatical errors throughout. I would suggest a very careful read/edit by a native English speaker/editor prior to publication.

Response: The language of our revised manuscript has been essentially improved by Professor Bruce Jaffee from UC Davis (<http://jaffeerevises.com/>; Invoice 1722).

Comment 24: Line 258: this is a disappointing way to end the paper. I don’t disagree with the statement, but it comes out of nowhere, and the context for it is not clear. I would either make it meaningful (explain) or omit it.

Response: As suggested, the statement has been removed from the revised manuscript.

REVIEWERS' COMMENTS:

Reviewer #3 (Remarks to the Author):

I appreciate the authors have made a very good faith effort to address my concerns and i am willing to support publication of the manuscript.